# A Theory of the Distortion-Perception Tradeoff in Wasserstein Space

**Dror Freirich**
Technion – Israel Institute of Technology
drorfrc@gmail.com

**Tomer Michaeli**
Technion – Israel Institute of Technology
tomer.m@ee.technion.ac.il

**Ron Meir**
Technion – Israel Institute of Technology
rmeir@ee.technion.ac.il

## Abstract

The lower the distortion of an estimator, the more the distribution of its outputs generally deviates from the distribution of the signals it attempts to estimate. This phenomenon, known as the perception-distortion tradeoff, has captured significant attention in image restoration, where it implies that fidelity to ground truth images comes at the expense of perceptual quality (deviation from statistics of natural images). However, despite the increasing popularity of performing comparisons on the perception-distortion plane, there remains an important open question: *what is the minimal distortion that can be achieved under a given perception constraint?* In this paper, we derive a closed form expression for this distortion-perception (DP) function for the mean squared-error (MSE) distortion and the Wasserstein-2 perception index. We prove that the DP function is always quadratic, regardless of the underlying distribution. This stems from the fact that estimators on the DP curve form a geodesic in Wasserstein space. In the Gaussian setting, we further provide a closed form expression for such estimators. For general distributions, we show how these estimators can be constructed from the estimators at the two extremes of the tradeoff: The global MSE minimizer, and a minimizer of the MSE under a perfect perceptual quality constraint. The latter can be obtained as a stochastic transformation of the former.

## 1 Introduction

Inverse problems that involve signal reconstruction from partial or noisy measurements, arise in numerous scientific domains. Examples range from medical imaging to tomography, microscopy, astronomy and audio enhancement. In many such problems it is desired to design an estimator that *(i)* has a small reconstruction error (low distortion), and *(ii)* outputs reconstructions that cannot be told apart from valid signals (good perceptual quality). Interestingly, however, it has been shown that the lower the average distortion of an estimator, the more the distribution of its outputs generally deviates from the distribution of the signals it attempts to estimate [4]. In other words, low distortion generally comes at the price of poor perceptual quality, and vice versa. This phenomenon, known as the *perception-distortion tradeoff*, has found particular interest in the image restoration domain (see Fig. 1), where algorithms are now commonly being evaluated using both distortion measures and perception indices [5].

Unfortunately, despite the increasing popularity of performing comparisons on the perception-distortion plane, the minimal distortion that can be achieved under a given perception constraint (red curve in Fig. 1) remains an open question. Blau and Michaeli [4] investigated several properties

35th Conference on Neural Information Processing Systems (NeurIPS 2021).

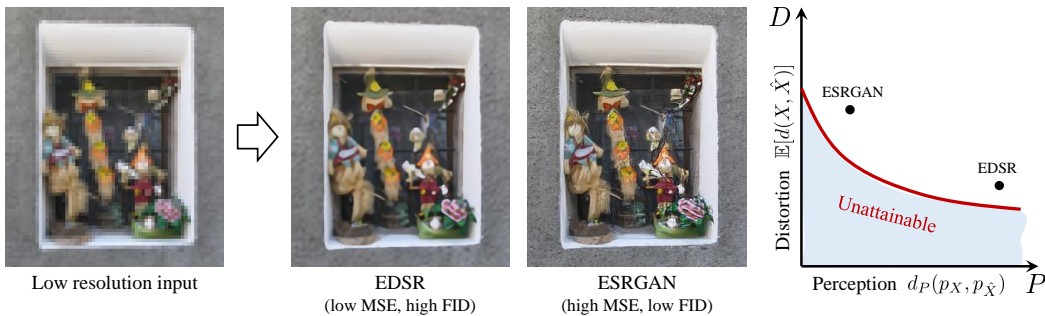

Figure 1: **Illustration of the distortion-perception tradeoff in super-resolution.** A low resolution image (left) is fed to two state-of-the-art super-resolution algorithms (middle). EDSR [14] achieves a low MSE distortion, but produces blurry reconstructions with high FID values [10]. ESRGAN [31] outputs photo-realistic recoveries with low FID, but its MSE is significantly higher. This is a result of the distortion-perception tradeoff (right). Namely, estimators cannot simulatenously achieve a low distortion and have their outputs distributed like the signals they are designed to estimate. In this paper, we derive a closed form expression for the distortion-perception function (red curve) for the MSE distortion and Wasserstein-2 perception index.

of this *distortion-perception function*, such as monotonicity and convexity. But beyond this rather general characterization, little is known about its precise nature. In this paper, we derive a closed form expression for the distortion-perception (DP) function for the special case where distortion is measured by mean squared-error (MSE) and perception is measured by the Wasserstein-2 distance between the probability laws of the estimate and the estimand.

Our main contributions are: *(i)* We prove that the DP function is *always* quadratic in the perception constraint $P$, regardless of the underlying distribution (Theorem 1). *(ii)* We show that it is possible to construct estimators on the DP curve from the estimators at the two extremes of the tradeoff (Theorem 3): The one that globally minimizes the MSE, and a minimizer of the MSE under a perfect perceptual quality constraint. The latter can be obtained as a stochastic transformation of the former. *(iii)* In the Gaussian setting, we further provide a closed form expression for optimal estimators and for the corresponding DP curve (Theorems 4 and 5). We show this Gaussian DP curve is a lower bound on the DP curve of any distribution having the same second order statistics. Finally, we illustrate our results, numerically and visually, in a super-resolution setting in Section 5. The proofs of all our theorems are provided in Appendix B.

Our theoretical results shed light on several topics that are subject to much practical activity. Particularly, many recent works adress the task of *diverse* perceptual image reconstruction, by employing randomization among possible restorations [15, 3, 22, 1]. Commonly, such works attempt to sample from the posterior distribution of natural images given the degraded input image. This is done, for example, using priors over image patches [7], conditional generative models [18, 21], or implicit priors induced by deep denoiser networks [11]. Theoretically, posterior sampling leads to perfect perceptual quality (the restored outputs are distributed like the prior). However, a fundamental question is whether this is optimal in terms of distortion. As we show in Section 3.1, posterior sampling is often not an optimal strategy, in the sense that there exist perfect perceptual quality estimators that achieve lower distortion.

Another topic of practical interest is the ability to *traverse the distortion-perception tradeoff* at test time, without having to train a different model for each working point. Recently, interpolation between distortion-oriented models and perception-oriented ones, has been suggested for this end. Existing methods include interpolation in pixel space [31] or in some latent space [26], interpolation between network weights [31, 32], and style transfer between a low-distortion reconstruction and a high perceptual quality one [6]. In light of this plethora of approaches, it is natural to ask which strategy is optimal. In Section 3.2 we show that for the MSE–Wasserstein-2 tradeoff, linear interpolation in pixel space leads to optimal estimators. We also discuss a geometric connection between interpolation and the fact that estimators on the DP curve form a geodesic in Wasserstein space.

## 2 Problem setting and preliminaries

### 2.1 The distortion-perception tradeoff

Let $X, Y$ be random vectors taking values in $\mathbb{R}^{n_x}$ and $\mathbb{R}^{n_y}$, respectively. We consider the problem of constructing an estimator $\hat{X}$ of $X$ based on $Y$. Namely, we are interested in determining a conditional distribution $p_{\hat{X}|Y}$ such that $\hat{X}$ constitutes a good estimate of $X$. For example, in the super-resolution setting shown in Fig. 1, $Y$ is the low resolution image (left), $X$ is the corresponding ground-truth high-resolution image (not shown), and $\hat{X}$ is a super-resolution reconstruction generated from $Y$ (e.g. the EDSR or ESRGAN estimators in the middle).

In many practical cases, the goodness of an estimator is associated with two factors: (i) the degree to which $\hat{X}$ is close to $X$ on average (low distortion), and (ii) the degree to which the distribution of $\hat{X}$ is close to that of $X$ (good perceptual quality). An important question, then, is *what is the minimal distortion that can be achieved under a given level of perceptual quality?* and *how can we construct estimators that achieve this lower bound?* In mathematical language, we are interested in analyzing the distortion-perception (DP) function (defined similarly to the perception-distortion function of [4])

$$D(P) = \min_{p_{\hat{X}|Y}} \left\{ \mathbb{E}[d(X, \hat{X})] \ : \ d_p(p_X, p_{\hat{X}}) \leq P \right\}. \tag{1}$$

Here, $d : \mathbb{R}^{n_x} \times \mathbb{R}^{n_x} \to \mathbb{R}^+ \cup \{0\}$ is some distortion criterion, $d_p(\cdot, \cdot)$ is some divergence between probability measures, and $p_{\hat{X}}$ is the probability measure on $\mathbb{R}^{n_x}$ induced by $p_{\hat{X}|Y}$ and $p_Y$. The expectation is taken w.r.t. the measure $p_{X\hat{X}}$ induced by $p_{\hat{X}|Y}$ and $p_{XY}$, where we assume that $\hat{X}$ is independent of $X$ given $Y$.

As discussed in [4], the function $D(P)$ is monotonically non-increasing and is convex whenever $d_p(\cdot, \cdot)$ is convex in its second argument (which is the case for most popular divergences). However, without further concrete assumptions on the distortion measure $d(\cdot, \cdot)$ and the perception index $d_p(\cdot, \cdot)$, little can be said about the precise nature of $D(P)$.

Here, we focus our attention on the squared-error distortion $d(x, \hat{x}) = \|x - \hat{x}\|^2$ and the Wasserstein-2 distance $d_p(p_X, p_{\hat{X}}) = W_2(p_X, p_{\hat{X}})$, with which (1) reads

$$D(P) = \min_{p_{\hat{X}|Y}} \left\{ \mathbb{E}[\|X - \hat{X}\|^2] \ : \ W_2(p_X, p_{\hat{X}}) \leq P \right\}. \tag{2}$$

We assume that all distributions have finite first and second moments. In addition, from Theorem 3 below it will follow that the minimum is indeed attained, so that (2) is well defined.

It is well known that the estimator minimizing the mean squared error (MSE) without any constraints, is given by $X^* = \mathbb{E}[X|Y]$. This implies that $D(P)$ monotonically decreases until $P$ reaches $P^* \triangleq W_2(p_X, p_{X^*})$, beyond which point $D(P)$ takes the constant value $D^* \triangleq \mathbb{E}[\|X - X^*\|^2]$. This is illustrated in Fig. 2. It is also known that $D(0) \leq 2D^*$ since the posterior sampling estimator $p_{\hat{X}|Y} = p_{X|Y}$ achieves $W_2(p_X, p_{\hat{X}}) = 0$ and $\mathbb{E}[\|X - \hat{X}\|^2] = 2D^*$ [4]. However, apart from these rather general properties, the precise shape of the DP curve has not been determined to date, and neither have the estimators that achieve the optimum in (2). This is our goal in this paper.

### 2.2 The Wasserstein and Gelbrich Distances

Before we present our main results, we briefly survey a few properties of the Wasserstein distance, mostly taken from [20]. The Wasserstein-$p$ ($p \geq 1$) distance between measures $\mu$ and $\gamma$ on a separable Banach space $\mathcal{X}$ with norm $\|\cdot\|$ is defined by

$$W_p^p(\mu, \gamma) \triangleq \inf \left\{ \mathbb{E}_{(U,V)\sim\nu}[\|U - V\|^p] \ : \ \nu \in \Pi(\mu, \gamma) \right\}, \tag{3}$$

where $\Pi(\mu, \gamma)$ is the set of all probabilities on $\mathcal{X} \times \mathcal{X}$ with marginals $\mu$ and $\gamma$. A joint probability $\nu$ achieving the optimum in (3) is often referred to as *optimal plan*. The Wasserstein space of probability measures is defined as

$$\mathcal{W}_p(\mathcal{X}) \triangleq \left\{ \gamma : \int_{\mathcal{X}} \|x\|^p d\gamma < \infty \right\},$$

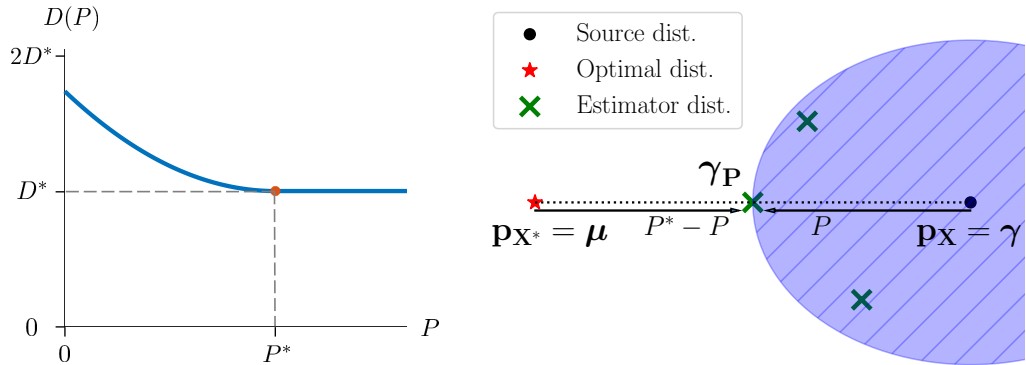

Figure 2: **The MSE–Wasserstein-2 tradeoff and the geometry of optimal estimators.** The left pane depicts the distortion-perception function for the MSE distortion and the Wasserstein-2 perception index. The minimal possible distortion, $D^*$, is achieved by the estimator $X^* = E[X|Y]$. The perception index attained by this estimator is $P^*$. At the other extreme of the tradeoff, we know that $D(0) \leq 2D^*$. The right pane shows the geometry of the distributions of optimal estimators in Wasserstein space. The minimal distortion $D(P)$ can be achieved by an estimator with distribution $\gamma_P$, which lies on a straight line (or geodesic) between $p_X$ and $p_{X^*}$. Its distance from the former is $W_2(p_X, \gamma_P) = P$ and its distance from the latter is $W_2(p_{X^*}, \gamma_P) = P^* - P$. Therefore, $D(P) = D^* + W_2^2(p_{X^*}, \gamma_P) = D^* + (P^* - P)^2$.

and $W_p$ constitutes a metric on $\mathcal{W}_p(\mathcal{X})$.

For any $(m_1, \Sigma_1), (m_2, \Sigma_2) \in \mathbb{R}^d \times \mathbb{S}_+^d$ (where $\mathbb{S}_+^d$ is the set of symmetric positive semidefinite matrices in $\mathbb{R}^{d \times d}$), the Gelbrich distance is defined as

$$G^2((m_1, \Sigma_1), (m_2, \Sigma_2)) \triangleq \|m_1 - m_2\|_2^2 + \text{Tr}\left\{\Sigma_1 + \Sigma_2 - 2\left(\Sigma_1^{\frac{1}{2}}\Sigma_2\Sigma_1^{\frac{1}{2}}\right)^{\frac{1}{2}}\right\}. \quad (4)$$

The root of a PSD matrix is always taken to be PSD. For any two probability measures $\mu_1, \mu_2$ on $\mathbb{R}^d$ with means and covariances $(m_1, \Sigma_1), (m_2, \Sigma_2)$, from [8, Thm. 2.1] we have that

$$W_2^2(\mu_1, \mu_2) \geq G^2((m_1, \Sigma_1), (m_2, \Sigma_2)). \quad (5)$$

When $\mu_1 = \mathcal{N}(m_1, \Sigma_1)$ and $\mu_2 = \mathcal{N}(m_2, \Sigma_2)$ are Gaussian distributions on $\mathbb{R}^d$, we have that $W_2(\mu_1, \mu_2) = G((m_1, \Sigma_1), (m_2, \Sigma_2))$. This equality is obvious for non-singular measures but is true for any two Gaussian distributions [20, p. 18]. If $\Sigma_1$ and $\Sigma_2$ are non-singular, then the distribution attaining the optimum in (3) corresponds to

$$U \sim \mathcal{N}(m_1, \Sigma_1), \quad V = m_2 + T_{1 \to 2}(U - m_1), \quad (6)$$

where

$$T_{1 \to 2} = \Sigma_1^{-\frac{1}{2}}\left(\Sigma_1^{\frac{1}{2}}\Sigma_2\Sigma_1^{\frac{1}{2}}\right)^{\frac{1}{2}}\Sigma_1^{-\frac{1}{2}} \quad (7)$$

is the optimal transformation pushing forward from $\mathcal{N}(0, \Sigma_1)$ to $\mathcal{N}(0, \Sigma_2)$ [12]. This transformation satisfies $\Sigma_2 = T_{1 \to 2}\Sigma_1 T_{1 \to 2}$. For a discussion on singular distributions, please see App. A.

## 3 Main results

### 3.1 The MSE–Wasserstein-2 tradeoff

The DP function (2) depends, of course, on the underlying joint probability $p_{XY}$ of the signal $X$ and measurements $Y$. Our first key result is that this dependence can be expressed solely in terms of $D^*$ and $P^*$. In other words, knowing the distortion and perception index attained by the minimum MSE estimator $X^*$, suffices for determining $D(P)$ for any $P$.

**Theorem 1** (The DP function). *The DP function* (2) *is given by*

$$D(P) = D^* + [(P^* - P)_+]^2, \tag{8}$$

*where* $(x)_+ = \max(0, x)$. *Furthermore, an estimator achieving perception index $P$ and distortion $D(P)$ can always be constructed by applying a (possibly stochastic) transformation to $X^*$.*

Theorem 1 is of practical importance because in many cases constructing an estimator that achieves a low MSE (*i.e.* an approximation of $X^*$) is a rather simple task. This is the case, for example, in image restoration with deep neural networks. There, it is common practice to train a network by minimizing its average squared error on a training set. Measuring the MSE of such a network on a large test set provides an approximation for $D^*$. We can also obtain an approximation of at least a lower bound on $P^*$ by estimating the second order statistics of $X$ and $X^*$. Specifically, recall that $P^*$ is lower bounded by the Gelbrich distance between $(m_X, \Sigma_X)$ and $(m_{X^*}, \Sigma_{X^*})$, which is given by $(G^*)^2 \triangleq \text{Tr}\{\Sigma_X + \Sigma_{X^*} - 2(\Sigma_X^{1/2}\Sigma_{X^*}\Sigma_X^{1/2})^{1/2}\}$ (see (5)). Given approximations for $D^*$ and $G^*$, we can approximate a lower bound on the DP function for any $P$,

$$D(P) \geq D^* + [(G^* - P)_+]^2. \tag{9}$$

The bound is attained when $X$ and $Y$ are jointly Gaussian.

**Uniqueness** A remark is in place regarding the uniqueness of an estimator achieving (8). As we discuss below, what defines an optimal estimator $\hat{X}$ is its joint distribution with $X^*$. This joint distribution may not be unique, in which case the optimal estimator is not unique. Moreover, even if $p_{\hat{X}X^*}$ is unique, the uniqueness of the estimator is not guaranteed because there may be different conditional distributions $p_{\hat{X}|Y}$ that lead to the same optimal $p_{\hat{X}X^*}$. In other words, given the optimal $p_{\hat{X}X^*}$, one can choose any joint probability $p_{\hat{X}YX^*}$ that has marginals $p_{\hat{X}X^*}$ and $p_{YX^*}$. One option is to take the estimator $\hat{X}$ to be a (possibly stochastic) transformation of $X^*$, namely $p_{\hat{X}|Y} = p_{\hat{X}|X^*}p_{X^*|Y}$. But there may be other options. In cases where either $Y$ or $\hat{X}$ are a deterministic transformation of $X^*$ (*e.g.* when $X^*$ has a density, or is an invertible function of $Y$), there is a unique joint distribution $p_{\hat{X}YX^*}$ with the given marginals [2, Lemma 5.3.2]. In this case, if $p_{\hat{X}X^*}$ is unique then so is the estimator $p_{\hat{X}|Y}$.

**Randomness** Under the settings of image restoration, many methods encourage diversity in their output by adding randomness [15, 3, 22]. In our setting, we may ask under what conditions there exists an optimal estimator $\hat{X}$ which is a deterministic function of $Y$. For example, when $p_Y = \delta_0$ but $X$ has some non-atomic distribution, it is clear that no deterministic function of $Y$ can attain perfect perceptual quality. It turns out that a sufficient condition for the optimal $\hat{X}$ to be a deterministic function of $Y$ is that $X^*$ have a density. We discuss this in App. B and explicitly illustrate it in the Gaussian case (see Sec. 3.3), where if $X^*$ has a non-singular covariance matrix then $\hat{X}$ is a deterministic function of $Y$.

**When is posterior sampling optimal?** Many recent image restoration methods attempt to produce diverse high perceptual quality reconstructions by sampling from the posterior distribution $p_{X|Y}$ [7, 18, 11]. As discussed in [4], the posterior sampling estimator attains a perception index of 0 (namely $W_2(p_X, p_{\hat{X}}) = 0$) and distortion $2D^*$. But an interesting question is: when is this strategy optimal? In other words, in what cases do we have that the DP function at $P = 0$ equals precisely $2D^*$ and is not strictly smaller? Note from the definition of the Wasserstein distance (3), that $(P^*)^2 = W_2^2(p_X, p_{X^*}) \leq \mathbb{E}[\|X - X^*\|^2] = D^*$. Using this in (8) shows that the DP function at $P = 0$ is upper bounded by

$$D(0) = D^* + (P^*)^2 \leq 2D^*, \tag{10}$$

and the upper bound is attained when $(P^*)^2 = D^*$. To see when this happens, observe that

$$\text{Tr}\left\{\Sigma_X + \Sigma_{X^*} - 2(\Sigma_X^{\frac{1}{2}}\Sigma_{X^*}\Sigma_X^{\frac{1}{2}})^{\frac{1}{2}}\right\} = (G^*)^2 \leq (P^*)^2 \leq D^* = \text{Tr}\{\Sigma_X - \Sigma_{X^*}\}. \tag{11}$$

We can see that when $\text{Tr}\{\Sigma_{X^*}\} = \text{Tr}\{(\Sigma_X^{1/2}\Sigma_{X^*}\Sigma_X^{1/2})^{1/2}\}$, the leftmost and rightmost sides become equal, and thus $(P^*)^2 = D^*$. To understand the meaning of this condition, let us focus on the case where $\Sigma_X$ and $\Sigma_{X^*}$ are jointly diagonalizable. This is a reasonable assumption for natural images,

where shift-invariance induces diagonalization by the Fourier basis [30]. In this case, the condition can be written in terms of the eigenvalues of the matrices, namely $\sum_i \lambda_i(\Sigma_{X^*}) = \sum_i \sqrt{\lambda_i(\Sigma_{X^*})\lambda_i(\Sigma_X)}$. This condition is satisfied when each $\lambda_i(\Sigma_{X^*})$ equals either $\lambda_i(\Sigma_X)$ or $0$. Namely, the $i$th eigenvalue of the error covariance of $X^*$, which is given by $\Sigma_X - \Sigma_{X^*}$, is either $\lambda_i(\Sigma_X)$ or $0$. We conclude that posterior sampling is optimal when there exists a subspace $\mathcal{S}$ spanned by some of the eigenvectors of $\Sigma_X$, such that the projection of $X$ onto $\mathcal{S}$ can be recovered from $Y$ with zero error, but the projection of $X$ onto $\mathcal{S}^\perp$ cannot be recovered at all (the optimal estimator is trivial). This is likely not the case in most practical scenarios. Therefore, it seems that *posterior sampling is often not optimal*. That is, posterior sampling can be improved upon in terms of MSE without any sacrifice in perceptual quality.

## 3.2 Optimal estimators

While Theorem 1 reveals the shape of the DP function, it does not provide a recipe for constructing optimal estimators on the DP tradeoff. We now discuss the nature of such estimators.

Our first observation is that since $\hat{X}$ is independent of $X$ given $Y$, its MSE can be decomposed as $\mathbb{E}[\|X - \hat{X}\|^2] = \mathbb{E}[\|X - X^*\|^2] + \mathbb{E}[\|X^* - \hat{X}\|^2]$ (see App. B). Therefore, the DP function (2) can be equivalently written as

$$D(P) = D^* + \min_{p_{\hat{X}|Y}} \left\{ \mathbb{E}[\|\hat{X} - X^*\|^2] \; : \; W_2(p_X, p_{\hat{X}}) \leq P \right\}. \tag{12}$$

Note that the objective in (12) depends on the MSE between $\hat{X}$ and $X^*$, so that we can perform the minimization on $p_{\hat{X}|X^*}$ rather than on $p_{\hat{X}|Y}$ (once we determine the optimal $p_{\hat{X}|X^*}$ we can construct a consistent $p_{\hat{X}|Y}$ as discussed above).

Now, let us start by examining the leftmost side of the curve $D(P)$, which corresponds to a perfect perceptual quality estimator (*i.e.* $P = 0$). In this case, the constraint becomes $p_{\hat{X}} = p_X$. Therefore,

$$D(0) = D^* + \min_{p_{\hat{X}X^*}} \left\{ \mathbb{E}[\|\hat{X} - X^*\|^2] \; : \; p_{\hat{X}X^*} \in \Pi(p_X, p_{X^*}) \right\}, \tag{13}$$

where $\Pi(p_X, p_{X^*})$ is the set of all probabilities on $\mathbb{R}^{n_x} \times \mathbb{R}^{n_x}$ with marginals $p_X, p_{X^*}$. One may readily recognize this as the optimization problem underlying the Wasserstein-2 distance between $p_X$ and $p_{X^*}$. This leads us to the following conclusion.

**Theorem 2** (Optimal estimator for $P = 0$). *Let $\hat{X}_0$ be an estimator achieving perception index $0$ and MSE $D(0)$. Then its joint distribution with $X^*$ attains the optimum in the definition of $W_2(p_X, p_{X^*})$. Namely, $p_{\hat{X}_0 X^*}$ is an optimal plan between $p_X$ and $p_{X^*}$.*

Having understood the estimator $\hat{X}_0$ at the leftmost end of the tradeoff, we now turn to study optimal estimators for arbitrary $P$. Interestingly, we can show that Problem (12) is equivalent to (see App. B)

$$D(P) = D^* + \min_{p_{\hat{X}}} \left\{ W_2^2(p_{\hat{X}}, p_{X^*}) \; : \; W_2(p_{\hat{X}}, p_X) \leq P \right\}. \tag{14}$$

Namely, an optimal $p_{\hat{X}}$ is closest to $p_{X^*}$ among all distributions within a ball of radius $P$ around $p_X$, as illustrated in Fig. 2. Moreover, $p_{\hat{X}X^*}$ is an optimal plan between $p_{\hat{X}}$ and $p_{X^*}$. As it turns out, this somewhat abstract viewpoint leads to a rather practical construction for $\hat{X}$ from the estimators $\hat{X}_0$ and $X^*$ at the two extremes of the tradeoff. Specifically, we have the following result.

**Theorem 3** (Optimal estimators for arbitrary $P$). *Let $\hat{X}_0$ be an estimator achieving perception index $0$ and MSE $D(0)$. Then for any $P \in [0, P^*]$, the estimator*

$$\hat{X}_P = \left(1 - \frac{P}{P^*}\right)\hat{X}_0 + \frac{P}{P^*}X^* \tag{15}$$

*is optimal for perception index $P$. Namely, it achieves perception index $P$ and distortion $D(P)$.*

Theorem 3 has important implications for perceptual signal restoration. For example, in the task of image super-resolution, there exist many deep network based methods that achieve a low MSE [14, 29, 25]. These provide an approximation for $X^*$. Moreover, there is an abundance of methods that achieve good perceptual quality at the price of a reasonable degradation in MSE (often by

incorporating a GAN-based loss) [13, 31, 24]. These constitute approximations for $\hat{X}_0$. However, achieving results that strike other prescribed balances between MSE and perceptual quality commonly require training a different model for each setting. Shoshan et al. [26] and Navarrete Michelini et al. [17] tried to address this difficulty by introducing new training techniques that allow traversing the distortion-perception tradeoff at test time. However, Theorem 3 shows that such specialized training methods are not required in our setting. Having a model that leads to low MSE and one that leads to good perceptual quality, it is possible to construct any other estimator on the DP tradeoff, by simply averaging the outputs of these two models with appropriate weights. We illustrate this in Sec. 5.

### 3.3 The Gaussian setting

When $X$ and $Y$ are jointly Gaussian, it is well known that the minimum MSE estimator $X^*$ is a linear function of the measurements $Y$. However, it is not *a-priori* clear whether all estimators along the DP tradeoff are linear in this case, and what kind of randomness they possess. As we now show, equipped with Theorem 3, we can obtain closed form expressions for optimal estimators for any $P$. For simplicity, we assume here that $X$ and $Y$ have zero means and that $\Sigma_X, \Sigma_Y \succ 0$.

It is instructive to start by considering the simple case, where $\Sigma_{X^*}$ is non-singular (in Theorem 4 below we address the more general case of a possibly singular $\Sigma_{X^*}$). It is well known that

$$X^* = \Sigma_{XY}\Sigma_Y^{-1}Y, \qquad \Sigma_{X^*} = \Sigma_{XY}\Sigma_Y^{-1}\Sigma_{YX}. \tag{16}$$

Now, since we assumed that $\Sigma_X, \Sigma_{X^*} \succ 0$, we have from Theorem 2 and (6),(7) that

$$\hat{X}_0 = \Sigma_{X^*}^{-\frac{1}{2}}\left(\Sigma_{X^*}^{\frac{1}{2}}\Sigma_X\Sigma_{X^*}^{\frac{1}{2}}\right)^{\frac{1}{2}}\Sigma_{X^*}^{-\frac{1}{2}}X^*. \tag{17}$$

Finally, we know that $P^* = G^*$, which is given by the left-hand side of (11). Substituting these expressions into (15), we obtain that an optimal estimator for perception $P \in [0, G^*]$ is given by

$$\hat{X}_P = \left(\left(1 - \frac{P}{G^*}\right)\Sigma_{X^*}^{-\frac{1}{2}}\left(\Sigma_{X^*}^{\frac{1}{2}}\Sigma_X\Sigma_{X^*}^{\frac{1}{2}}\right)^{\frac{1}{2}}\Sigma_{X^*}^{-\frac{1}{2}} + \frac{P}{G^*}I\right)\Sigma_{XY}\Sigma_Y^{-1}Y. \tag{18}$$

As can be seen, this optimal estimator is a deterministic linear transformation of $Y$ for any $P$.

The setting just described does not cover the case where $Y$ is of lower dimensionality than $X$ because in that case $\Sigma_{X^*}$ is necessarily singular (it is a $n_x \times n_x$ matrix of rank at most $n_y$; see (16)). In this case, any deterministic linear function of $Y$ would result in an estimator $\hat{X}$ with a rank-$n_y$ covariance. Obviously, the distribution of such an estimator cannot be arbitrarily close to that of $X$, whose covariance has rank $n_x$. What is the optimal estimator in this more general setting, then?

**Theorem 4** (Optimal estimators in the Gaussian case). *Assume $X$ and $Y$ are zero-mean jointly Gaussian random vectors with $\Sigma_X, \Sigma_Y \succ 0$. Denote $T^* \triangleq T_{p_X \to p_{X^*}} = \Sigma_X^{-1/2}(\Sigma_X^{1/2}\Sigma_{X^*}\Sigma_X^{1/2})^{1/2}\Sigma_X^{-1/2}$. Then for any $P \in [0, G^*]$, an estimator with perception index $P$ and MSE $D(P)$ can be constructed as*

$$\hat{X}_P = \left(\left(1 - \frac{P}{G^*}\right)\Sigma_X^{\frac{1}{2}}\left(\Sigma_X^{\frac{1}{2}}\Sigma_{X^*}\Sigma_X^{\frac{1}{2}}\right)^{\frac{1}{2}}\Sigma_X^{-\frac{1}{2}}\Sigma_{X^*}^{\dagger} + \frac{P}{G^*}I\right)\Sigma_{XY}\Sigma_Y^{-1}Y + \left(1 - \frac{P}{G^*}\right)W, \tag{19}$$

*where $W$ is a zero-mean Gaussian noise with covariance $\Sigma_W = \Sigma_X^{1/2}(I - \Sigma_X^{1/2}T^*\Sigma_{X^*}^{\dagger}T^*\Sigma_X^{1/2})\Sigma_X^{1/2}$, which is independent of $Y, X$, and $\Sigma_{X^*}^{\dagger}$ is the pseudo-inverse of $\Sigma_{X^*}$.*

Note that in this case, we indeed have a random noise component that shapes the covariance of $\hat{X}_P$ to become closer to $\Sigma_X$ as $P$ gets closer to 0. It can be shown (see App. B) that when $\Sigma_{X^*}$ is invertible, $\Sigma_W = 0$ and (19) reduces to (18). Also note that, as in (18), the dependence of $\hat{X}_P$ on $Y$ in (19) is only through $X^* = \Sigma_{XY}\Sigma_Y^{-1}Y$.

As mentioned in Sec. 3.1, the optimal estimator is generally not unique. Interestingly, in the Gaussian setting we can explicitly characterize a *set* of optimal estimators.

**Theorem 5** (A set of optimal estimators in the Gaussian case). *Consider the setting of Theorem 4. Let $\Sigma_{\hat{X}_0Y} \in \mathbb{R}^{n_x \times n_y}$ satisfy*

$$\Sigma_{\hat{X}_0Y}\Sigma_Y^{-1}\Sigma_{YX} = \Sigma_X^{\frac{1}{2}}(\Sigma_X^{\frac{1}{2}}\Sigma_{X^*}\Sigma_X^{\frac{1}{2}})^{\frac{1}{2}}\Sigma_X^{-\frac{1}{2}}, \tag{20}$$

*and $W_0$ be a zero-mean Gaussian noise with covariance*

$$\Sigma_{W_0} = \Sigma_X - \Sigma_{\hat{X}_0 Y} \Sigma_Y^{-1} \Sigma_{\hat{X}_0 Y}^T \succeq 0 \tag{21}$$

*that is independent of $X, Y$. Then, for any $P \in [0, G^*]$, an optimal estimator with perception index $P$ can be obtained by*

$$\hat{X}_P = \left( \left( 1 - \frac{P}{G^*} \right) \Sigma_{\hat{X}_0 Y} + \frac{P}{G^*} \Sigma_{XY} \right) \Sigma_Y^{-1} Y + \left( 1 - \frac{P}{G^*} \right) W_0. \tag{22}$$

*The estimator given in (19) is one solution to (20)-(21), but is generally not unique.*

### 3.4 A Comment on the MSE–Wasserstein-$p$ tradeoff

While our results concern the $\text{MSE} - W_2$ tradeoff, they can be used to draw conclusions regarding the DP tradeoff with other divergences. In particular, (8) constitutes a lower bound on the MSE-Wasserstein-$p$ tradeoff for any $p \geq 2$. Furthermore, we can show that the $\text{MSE} - W_1$ DP function is lower bounded by $D^* + [(P_1^* - P)_+]^2$, where $P_1^* \triangleq W_1(p_X, p_{X^*})$.

Note that at the point $P = 0$, the DP function coincides with (8) for any plausible divergence. For a detailed discussion, we kindly refer the reader to the Appendix.

## 4 A geometric perspective on the distortion-perception tradeoff

In this section we provide a geometric point of view on our main results. Specifically, we show that the results of Theorems 1 and 3 are a consequence of a more general geometric property of the space $\mathcal{W}_2(\mathbb{R}^{n_x})$. In the Gaussian case, this is simplified to a geometry of covariance matrices.

Recall from (14) that the optimal $p_{\hat{X}}$ is the one closest to $p_{X^*}$ (in terms of Wasserstein distance) among all measures at a distance $P$ from $p_X$. This implies that to determine $p_{\hat{X}}$, we should traverse the geodesic between $p_{X^*}$ and $p_X$ until reaching a distance of $P$ from $p_X$. Furthermore, $p_{\hat{X} X^*}$ should be the optimal plan between $p_{\hat{X}}$ and $p_{X^*}$. Interestingly, geodesics in Wasserstein spaces take a particularly simple form, and their explicit construction also turns out to satisfy the latter requirement. Specifically, let $\gamma, \mu$ be measures in $\mathcal{W}_2(\mathbb{R}^d)$, let $\nu \in \Pi(\gamma, \mu)$ be an optimal plan attaining $W_2(\gamma, \mu)$, and let $\pi_i$ denote the projection $\pi_i : \mathbb{R}^d \times \mathbb{R}^d \to \mathbb{R}^d$ such that $\pi_i((x_1, x_2)) = x_i, \ i = 1, 2$. Then, the curve

$$\gamma_t \triangleq [(1 - t)\pi_1 + t\pi_2] \# \nu, \quad t \in [0, 1] \tag{23}$$

is a constant-speed geodesic from $\gamma$ to $\mu$ in $\mathcal{W}_2(\mathbb{R}^d)$ [2], where $\#$ is the push-forward operation[1]. Particularly,

$$W_2(\gamma_t, \gamma_s) = |t - s| W_2(\gamma, \mu), \tag{24}$$

and it follows that $W_2(\gamma_t, \gamma) = t W_2(\gamma, \mu)$ and $W_2(\gamma_t, \mu) = (1 - t) W_2(\gamma, \mu)$. Furthermore, if $\gamma_t, t \in [0, 1]$ is a constant-speed geodesic with $\gamma_0 = \gamma, \gamma_1 = \mu$, then the optimal plans between $\gamma, \gamma_t$ and between $\gamma_t, \mu$ are given by

$$[\pi_1, (1 - t)\pi_1 + t\pi_2] \# \nu, \quad [(1 - t)\pi_1 + t\pi_2, \pi_2] \# \nu, \tag{25}$$

respectively, where $\nu \in \Pi(\gamma, \mu)$ is some optimal plan. Applying (23) to $(\hat{X}_0, X^*) \sim \nu$ with $t = P/P^*$, we obtain (15), where we show that the obtained estimator achieves $\mathbb{E}[\|\hat{X}_P - X^*\|^2] = (1 - t)^2 W_2^2(p_X, p_{X^*})$. This explains the result of Theorem 3.

It is worth mentioning that this geometric interpretation is simplified under some common settings. For example, when $\gamma$ is absolutely continuous (w.r.t. the Lebesgue measure), we have a measurable map $T_{\gamma \to \mu}$ which is the solution to the optimal transport problem with the quadratic cost [20, Thm 1.6.2, p.16]. The geodesic (23) then takes the form

$$\gamma_t = [Id + t(T_{\gamma \to \mu} - Id)] \# \gamma, \quad t \in [0, 1]. \tag{26}$$

Therefore, in our setting, if $\gamma = p_{X^*}$ has a density, then we can obtain $\hat{X}_P$ by the deterministic transformation $[X^* + (1 - \frac{P}{P^*})(T_{p_{X^*} \to p_X}(X^*) - X^*)]$ (see Remark about randomness in Sec. 3.1).

---

[1] For measures $\gamma, \mu$ on $\mathcal{X}, \mathcal{Y}$, we say that a measurable transform $T : \mathcal{X} \to \mathcal{Y}$ pushes $\gamma$ forward to $\mu$ (denoted $T \# \gamma = \mu$) iff $\gamma(T^{-1}(B)) = \mu(B)$ for any measurable $B \subseteq \mathcal{Y}$.

Further simplification arises when $\gamma, \mu$ are centered non-singular Gaussian measures, in which case $T_{\gamma \to \mu}$ is the linear and symmetric transformation (7). Then, $\gamma_t$ is a Gaussian measure with covariance $\Sigma_{\gamma_t} = T_t \Sigma_\gamma T_t$, where $T_t \triangleq [I + t(T_{\gamma \to \mu} - I)]$. Therefore, in the Gaussian case, the shortest path (23) between distributions is reduced to a trajectory in the geometry of covariance matrices induced by the Gelbrich distance [27]. If additionally $\Sigma_\gamma$ and $\Sigma_\mu$ commute, then the Gelbrich distance is further reduced to the $\ell^2$-distance between matrices, as we discuss in App. D.

## 5    Numerical illustration

We now experimentally illustrate the results of Theorems 1 and 3. We compute distortion and perception indices for 13 super resolution algorithms in a $4\times$ magnification task on the BSD100 dataset[2] [16]. We then demonstrate the efficiency of approximating the lower bound (9) on the distortion $D(P)$, and of the estimators suggested in (15), in this practical setting. The evaluated algorithms include EDSR [14], ESRGAN [31], SinGAN [24], ZSSR [25], DIP [29], SRResNet variants which optimize MSE and VGG$_{2,2}$, SRGAN variants which optimize MSE, VGG$_{2,2}$ and VGG$_{5,4}$ in addition to an adversarial loss [13], ENet [23] ("PAT" and "E" variants), and the stochastic explorable SR method of [3] (ExpSR). Low resolution images were obtained by $4\times$ downsampling of BSD100 images using a bicubic kernel.

In Fig. 3 we plot each method on the distortion-perception plane. In the left pane, we consider natural (and reconstructed) images to be stationary random sources, and use $9 \times 9$ patches (totally $1.6 \times 10^6$ patches) to empirically estimate the mean and covariance matrix for the ground-truth images, and for the reconstructions produced by each method. We then use the estimated Gelbrich distances (4) between the patch distribution of each method and that of ground-truth images, as a perceptual quality index. Recall this is a lower bound on the Wasserstein distance. In the right pane of Fig. 3, we measure perception using the popular FID index [10], which relies on the Fréchet distance between deep feature distributions of ground-truth and reconstructed images (assuming they are normally distributed). FID is known to correlate well with visual quality, and while it is not directly related to our theory, we can see a qualitatively similar behavior to that depicted in the left pane.

We consider the EDSR method [14] to constitute a good approximation for the minimum MSE estimator $X^*$ since it achieves the lowest MSE among the evaluated methods. We therefore estimate the lower bound (9) as

$$\hat{D}(P) = D_{\text{EDSR}} + \left[(P_{\text{EDSR}} - P)_+\right]^2,$$

where $D_{\text{EDSR}}$ is the MSE of EDSR, and $P_{\text{EDSR}}$ is the estimated Gelbrich distance between EDSR reconstructions and ground-truth images. Note the unoccupied region under the estimated curve in Fig. 3, which is indeed unattainable according to the theory.

The figure also shows 9 estimators $\hat{X}_t$, which we construct by interpolation between EDSR and ESRGAN, $\hat{X}_t = t X_{\text{EDSR}} + (1 - t) X_{\text{ESRGAN}}$ with $t \in [0, 1]$. We observe that estimators constructed using these two extreme points are closer to the optimal DP tradeoff than the other evaluated methods. This is true both for the Gelbrich perception index and for FID. In Fig. 4 we present a visual comparison between SRGAN-VGG$_{2,2}$ [13] and our interpolated estimator $\hat{X}_{0.12}$. Both achieve roughly the same RMSE distortion (18.08 for SRGAN, 18.14 for $\hat{X}_{0.12}$), but our estimator achieves a lower perception index. Namely, by using interpolation, we manage to achieve improvement in perceptual quality, without degradation in distortion. The improvement in visual quality is also apparent in the figure. Additional visual comparisons including more points along the DP curve and ground-truth images can be found in the Appendix.

## 6    Conclusion

In this paper we provide a full characterization of the distortion-perception tradeoff for the MSE distortion and the Wasserstein-2 perception index. We show that optimal estimators are obtained by interpolation between the minimum MSE estimator and an optimal perfect perceptual quality estimator. In the Gaussian case, we explicitly formulate these estimators. To the best of our knowledge, this is the first work to derive such closed-form expressions. Our work paves the way towards fully

---

[2]All codes are freely available and provided by the authors.

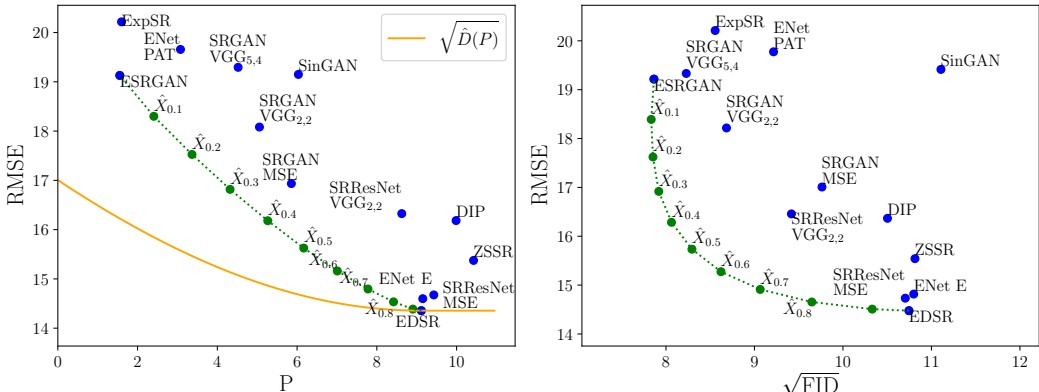

Figure 3: **Evaluation of SR algorithms**. We plot 13 algorithms (blue) on the Distortion-Perception plane. In the left pane, perception is measured using the Gelbrich distance between empirical means and covariances of patches from the ground-truth images and the reconstructed images. In the right pane, we measure perception using FID. In orange is the estimated lower bound (9), where we consider EDSR to be the global minimizer $X^*$. Note the unoccupied region under the estimated curve, which is unattainable. We also plot 9 estimators $\hat{X}_t$ (Green) created by interpolation between EDSR and ESRGAN reconstructions, using different relative weights $t$. Note that estimators constructed using these two extreme estimators are closer to the optimal DP curve than the compared methods.

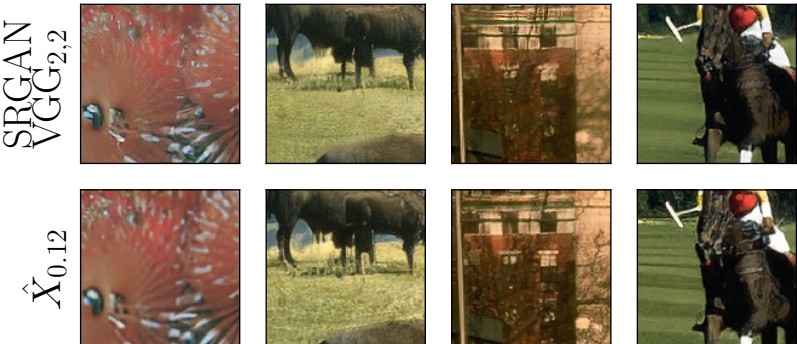

Figure 4: **A visual comparison between estimators with approximately the same MSE**. Upper: SRGAN-VGG$_{2,2}$. Lower: $\hat{X}_{0.12}$, an interpolation between EDSR and ESRGAN using $t = 0.12$. Observe the improvement in perceptual quality, without any significant degradation in distortion.

understanding the DP tradeoff under more general distortions and perceptual criteria, and bridging between fidelity and visual quality at test-time, without training different models.

**Broader impact** Synthesis of photo-realistic visual contents may raise concerns of inappropriate and malicious use. This is true for image generation in general (e.g. with GANs), but to some extent also for image restoration tasks like super-resolution. However, even without malicious intent, the outputs of a high perceptual quality algorithm can often not be very close to the ground truth images. In this paper we quantify this effect, by studying the best similarity (lowest distortion) one can hope to achieve with an algorithm having a prescribed level of perceptual quality.

**Acknoledgments** This work was partially supported by grants 451/17 and 852/17 from the Israel Science Foundation, by the Ollendorff Center of the Viterbi Faculty of Electrical and Computer Engineering at the Technion, and by the Skillman chair in biomedical sciences.

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
