# A Theory of the Distortion-Perception Tradeoff in Wasserstein Space - Supplementary Material

In Appendix A we present the distortion-perception tradeoff in general metric spaces. We formulate the problem of finding a perfect perceptual quality estimator as an optimal transportation problem, and extend some of the background provided in Sec. 2. In Appendix B we provide detailed proofs of the results appearing in the paper. In Appendix C we discuss the implications of our results on the DP tradeoff with divergences other than the Wasserstein-2. Appendix D examines settings where covariance matrices commute. In Appendix E we discuss the details of the numerical illustrations of Sec. 5 and provide additional visual results. Appendix F summarizes the results in the paper.

## A    Background and extensions

### A.1    The distortion-perception function

In Sec. 2 of the main text we presented the setting of Euclidean space for simplicity. For the sake of completeness, we present here a more general setup.

Let $X, Y$ be random variables on separable metric spaces $\mathcal{X}, \mathcal{Y}$, with joint probability $p_{X,Y}$ on $\mathcal{X} \times \mathcal{Y}$. Given a distortion function $d : \mathcal{X} \times \mathcal{X} \to \mathbb{R}^+ \cup \{0\}$, we aim to find an estimator $\hat{X} \in \mathcal{X}$ defined by a conditional distribution $p_{\hat{X}|Y}$ (which induces a marginal distribution $p_{\hat{X}}$), minimizing the expectation $\mathbb{E}[d(X, \hat{X})]$ under the constraint $d_p(p_X, p_{\hat{X}}) \leq P$. Here, $d_p$ is some divergence between probability measures. We further assume the Markov relation $X \to Y \to \hat{X}$, i.e. $X, \hat{X}$ are independent given $Y$. Similarly to Blau and Michaeli [4] we define the distortion-perception function

$$D(P) = \min_{p_{\hat{X}|Y}} \left\{ \mathbb{E}[d(X, \hat{X})] \; : \; d_p(p_X, p_{\hat{X}}) \leq P \right\}. \tag{27}$$

The expectation is taken w.r.t. the joint probability $p_{\hat{X}YX}$ induced by $p_{\hat{X}|Y}$ and $p_{XY}$, where $\hat{X}$ and $X$ are independent given $Y$. We can write (27) as

$$D(P) = \min_{p_{\hat{X}|Y}} \left\{ J(p_{\hat{X}|Y}) \; : \; d_p(p_X, p_{\hat{X}}) \leq P \right\}, \tag{28}$$

where we defined $J(p_{\hat{X}|Y}) \triangleq \mathbb{E}_{p_{\hat{X}YX}}[d(X, \hat{X})]$. This objective can be written as

$$J(p_{\hat{X}|Y}) = \mathbb{E}_{p_{\hat{X}YX}} \mathbb{E}[d(X, \hat{X})|Y, \hat{X}]. \tag{29}$$

Let us define the cost function

$$\begin{aligned} \rho(\hat{x}, y) &\triangleq \mathbb{E}[d(X, \hat{X})|Y = y, \hat{X} = \hat{x}] \\ &= \mathbb{E}[d(X, \hat{x})|Y = y], \end{aligned} \tag{30}$$

where we used the fact that $X$ is independent of $\hat{X}$ given $Y$. Then we have that the objective (29) boils down to $J(p_{\hat{X}|Y}) = \mathbb{E}_{p_{\hat{X}Y}} \rho(\hat{X}, Y)$.

The problem of finding a *perfect* perceptual quality estimator can be now written as an optimal transport problem

$$D(P = 0) = \min_{p_{\hat{X}|\tilde{Y}}} \mathbb{E}_{p_{\hat{X}\tilde{Y}}} \rho(\hat{X}, \tilde{Y}) \quad \text{s.t.} \; p_{\hat{X}} = p_X, p_{\tilde{Y}} = p_Y.$$

In the setting where $\mathcal{X}, \mathcal{Y}$ are Euclidean spaces, considering the MSE distortion $d(x, \hat{x}) = \|x - \hat{x}\|^2$, we write

$$\begin{aligned} \rho(\hat{x}, y) &= \mathbb{E}\left[\|X - \hat{X}\|^2 | Y = y, \hat{X} = \hat{x}\right] \\ &= \mathbb{E}\left[\|X - \hat{x}\|^2 | Y = y\right] \\ &= \mathbb{E}\left[\|X\|^2 | Y = y\right] - 2\hat{x}^T \mathbb{E}[X|Y = y] + \|\hat{x}\|^2 \\ &= \mathbb{E}\left[\|X - X^*\|^2 | Y = y\right] + \left\{ \mathbb{E}\left[\|X^*\|^2 | Y = y\right] - 2\hat{x}^T \mathbb{E}[X|Y = y] + \|\hat{x}\|^2 \right\} \end{aligned}$$

and we have

$$J(p_{\hat{X}|Y}) = \mathbb{E}_{p_{\hat{X}YX}}\rho(\hat{X}, Y) = \mathbb{E}_{p_{YX}}\mathbb{E}\left[\|X - X^*\|^2|Y\right] + \mathbb{E}_{p_{\hat{X}Y}}\mathbb{E}\left[\|\hat{X} - X^*\|^2|Y, \hat{X}\right]$$

$$= D^* + \mathbb{E}_{p_{\hat{X}Y}}\left[\|\hat{X} - X^*\|^2\right].$$

## A.2 The optimal transportation problem

Assume $\mathcal{X}, \mathcal{Y}$ are Radon spaces [2]. Let $\rho : \mathcal{X} \times \mathcal{Y} \to \mathbb{R}$ be a non-negative Borel cost function, and let $q^{(x)}, p^{(y)}$ be probability measures on $\mathcal{X}, \mathcal{Y}$ respectively. The optimal transport problem is then given in the following formulations.

In the *Monge* formulation, we search for an optimal transformation, often referred to as an *optimal map*, $T : \mathcal{Y} \to \mathcal{X}$ minimizing

$$\mathbb{E}\rho(T(Y), Y), \text{ s.t. } Y \sim q^{(y)}, T(Y) \sim q^{(x)}. \tag{31}$$

Note that the Monge problem seeks for a deterministic map, and might not have a solution.

In the *Kantorovich* formulation, we wish to find a probability measure $q = q_{XY}$ on $\mathcal{X} \times \mathcal{Y}$, minimizing

$$\mathbb{E}_q\rho(X, Y), \text{ s.t. } q \in \Pi(q^{(x)}, p^{(y)}), \tag{32}$$

where $\Pi$ is the set of probabilities on $\mathcal{X} \times \mathcal{Y}$ with marginals $q^{(x)}, p^{(y)}$. A probability minimizing (32) is called an *optimal plan*, and we denote $q \in \Pi_o(q^{(x)}, p^{(y)})$. Note that when $\rho(x, y) = d^p(x, y)$ and $d(x, y)$ is a metric, taking inf over (32) yields the Wasserstein distance $W_p^p(q^{(x)}, p^{(y)})$ induced by $d(x, y)$.

In the case where $\mathcal{X} = \mathcal{Y} = \mathbb{R}^d$ and $\rho(x, y) = \|x - y\|^2$ is the quadratic cost (and we assume $q^{(x)}, p^{(y)}$ have finite first and second moments), there exists an optimal plan minimizing (32). If $p^{(y)}$ is absolutely continuous (w.r.t Lebesgue measure), this plan is given by an optimal map which is the unique solution to (31) [20, p.5,16].

## A.3 Optimal maps between Gaussian measures

When $\mu_1 = \mathcal{N}(m_1, \Sigma_1)$ and $\mu_2 = \mathcal{N}(m_2, \Sigma_2)$ are Gaussian distributions on $\mathbb{R}^d$, we have that

$$W_2^2(\mu_1, \mu_2) = \|m_1 - m_2\|_2^2 + \text{Tr}\left\{\Sigma_1 + \Sigma_2 - 2\left(\Sigma_1^{\frac{1}{2}}\Sigma_2\Sigma_1^{\frac{1}{2}}\right)^{\frac{1}{2}}\right\}. \tag{33}$$

If $\Sigma_1$ and $\Sigma_2$ are non-singular, then the distribution attaining the optimum in (3) corresponds to

$$U \sim \mathcal{N}(m_1, \Sigma_1), \quad V = m_2 + T_{1\to2}(U - m_1), \tag{34}$$

where

$$T_{1\to2} = \Sigma_1^{-\frac{1}{2}}\left(\Sigma_1^{\frac{1}{2}}\Sigma_2\Sigma_1^{\frac{1}{2}}\right)^{\frac{1}{2}}\Sigma_1^{-\frac{1}{2}} \tag{35}$$

is the optimal transformation pushing forward from $\mathcal{N}(0, \Sigma_1)$ to $\mathcal{N}(0, \Sigma_2)$ [12]. This transformation satisfies $\Sigma_2 = T_{1\to2}\Sigma_1 T_{1\to2}$.

When distributions are singular, we have the following.

**Lemma 1.** *[33, Theorem 3] Let $\mu$ and $\nu$ be two centered Gaussian measures defined on $\mathbb{R}^n$. Let $P_\mu$ be the projection matrix onto $\text{Im}\{\Sigma_\mu\}$. Then the optimal transport map $T_{\mu\to P_\mu\#\nu}$ from $\mu$ to $P_\mu\#\nu$ is linear and self-adjoint, and can be written as*

$$T_{\mu\to P_\mu\#\nu} = (\Sigma_\mu^{1/2})^\dagger(\Sigma_\mu^{1/2}\Sigma_\nu\Sigma_\mu^{1/2})^{1/2}(\Sigma_\mu^{1/2})^\dagger.$$

*In the case $\text{Im}\{\Sigma_\nu\} \subseteq \text{Im}\{\Sigma_\mu\}$ we have $P_\mu\#\nu = \nu$, hence $T_{\mu\to\nu} = T_{\mu\to P_\mu\#\nu}$ is the optimal transport map from $\mu$ to $\nu$, even where measures are singular.*

# B  Proof of main results

In this Section we provide proofs of the main results of this paper. In lemmas 2 and 3 we present some alternative representations for $D(P)$. In Lemma 4 we obtain a lower bound on $D(P)$. We then prove Theorem 3 (via a more general result given by Lemma 5), where the lower bound of Lemma 4 is attained. Equipped with Theorem 3, we prove Theorem 1 which is the main result of our paper.

## B.1  Relations between $D(P)$ and $X^*$

In this section we relate the distortion-perception function $D(P)$ given in (2) to the estimator $X^* = \mathbb{E}\left[X|Y\right]$. Recall that $D^* = \mathbb{E}\left[\|X - X^*\|^2\right]$ and $P^* = W_2(p_X, p_{X^*})$.

**Lemma 2.** *If $\hat{X}$ is independent of $X$ given $Y$, then its MSE can be decomposed as* $\mathbb{E}[\|X - \hat{X}\|^2] = \mathbb{E}[\|X - X^*\|^2 + \mathbb{E}[\|X^* - \hat{X}\|^2]$ *and hence*

$$D(P) = D^* + \min_{p_{\hat{X}|Y}} \left\{ \mathbb{E}_{p_{\hat{X}Y}} \left[\|\hat{X} - X^*\|^2\right] \; : \; W_2(p_{\hat{X}}, p_X) \leq P \right\}. \tag{36}$$

*Proof.* For any estimator we can write the MSE

$$\mathbb{E}\left[\|X - \hat{X}\|^2\right] = \mathbb{E}\left[|X - X^*\|^2\right] + \mathbb{E}\left[\|\hat{X} - X^*\|^2\right] - 2\mathbb{E}\left[(X - X^*)^T(\hat{X} - X^*)\right]. \tag{37}$$

Since in our case $\hat{X}$ is independent of $X$ given $Y$, we show that the third term vanishes.

$$\begin{aligned}
\mathbb{E}\left[(X - X^*)^T(\hat{X} - X^*)\right] &= \mathbb{E}\left[\mathbb{E}(X - X^*)^T(\hat{X} - X^*)|Y\right] \\
&= \mathbb{E}\left[\underbrace{\mathbb{E}\left[(X - X^*)^T|Y\right]}_{=0}\left[\mathbb{E}(\hat{X} - X^*)|Y\right]\right] = 0.
\end{aligned}$$

Since $X^*$ is a deterministic function of $Y$, $D^* = \mathbb{E}\left[\|X - X^*\|^2\right]$ is a property of the problem, and does not depend on the choice of $p_{\hat{X}|Y}$, which, in view of (37) completes the proof. $\square$

Next, we express $D(P)$ in terms of the Wasserstein distance between $p_{\hat{X}}$ and $p_{X^*}$.

**Lemma 3** (Eq. (14))**.**

$$D(P) = D^* + \min_{p_{\hat{X}}} \left\{ W_2^2(p_{\hat{X}}, p_{X^*}) \; : \; W_2(p_{\hat{X}}, p_X) \leq P \right\}. \tag{38}$$

*Proof.* Denote $W_2^2(\mathcal{B}_P, p_{X^*}) = \min_{p_{\hat{X}}:W_2(p_{\hat{X}}, p_X) \leq P} W_2^2(p_{\hat{X}}, p_{X^*})$, where $\mathcal{B}_P$ is the ball of radius $P$ around $p_X$ in Wasserstein space.

From Lemma 2 we have

$$D(P) = D^* + \min_{p_{\hat{X}|Y}:W_2(p_{\hat{X}}, p_X) \leq P} \mathbb{E}_{p_{\hat{X}Y}} \left[\|\hat{X} - X^*\|^2\right]. \tag{39}$$

For every $p_{\hat{X}|Y}$ whose marginal attains $W_2(p_{\hat{X}}, p_X) \leq P$ we have,

$$\begin{aligned}
\mathbb{E}_{p_{\hat{X}Y}} \left[\|\hat{X} - X^*\|^2\right] &\geq \inf_{q \in \Pi(p_{\hat{X}}, p_{X^*})} \mathbb{E}_q \left[\|\hat{X} - X^*\|^2\right] \\
&= W_2^2(p_{\hat{X}}, p_{X^*}) \\
&\geq \min_{p_{\hat{X}}:W_2(p_{\hat{X}}, p_X) \leq P} W_2^2(p_{\hat{X}}, p_{X^*}),
\end{aligned}$$

which leads to $D(P) \geq D^* + W_2^2(\mathcal{B}_P, p_{X^*})$.

Conversely, given $p_{\hat{X}}$ such that $W_2(p_{\hat{X}}, p_X) \leq P$, we have an optimal plan $p_{\hat{X}X^*}$ achieving $W_2(p_{\hat{X}}, p_{X^*})$. Once we determine the optimal plan $p_{\hat{X}X^*}$ with marginal $p_{\hat{X}}$, we have an estimator $\hat{X}$ given by $p_{\hat{X}|Y}$ achieving $\mathbb{E}_{p_{\hat{X}Y}} \left[\|\hat{X} - X^*\|^2\right] = W_2^2(p_{\hat{X}}, p_{X^*})$ (for the connection between the

optimal plan $p_{\hat{X}X^*}$ and the choice of a consistent $p_{\hat{X}|Y}$, see Remark about uniqueness in Sec. 3.1).
We then have

$$\min_{p_{\hat{X}|Y}:W_2(p_{\hat{X}},p_X)\leq P} \mathbb{E}_{p_{\hat{X}Y}}\left[\|\hat{X}-X^*\|^2\right] \leq \mathbb{E}_{p_{\hat{X}Y}}\left[\|\hat{X}-X^*\|^2\right] = W_2^2(p_{\hat{X}},p_{X^*}).$$

Taking the minimum over $p_{\hat{X}}$ yields $D(P) \leq D^* + W_2^2(\mathcal{B}_P, p_{X^*})$. Combining the upper and lower bounds, we obtain the desired result. □

For the proof of Theorem 3, we first prove the following

**Lemma 4.** $D(P) \geq D^* + [(P^* - P)_+]^2$.

*Proof.* For every estimator satisfying $W_2(p_{\hat{X}}, p_X) \leq P$, we have from the triangle inequality

$$P^* = W_2(p_X, p_{X^*}) \leq W_2(p_{\hat{X}}, p_{X^*}) + W_2(p_{\hat{X}}, p_X) \leq W_2(p_{\hat{X}}, p_{X^*}) + P, \tag{40}$$

yielding

$$\mathbb{E}\left[\|X-\hat{X}\|^2\right] = \mathbb{E}\left[\|X-X^*\|^2\right] + \mathbb{E}\left[\|\hat{X}-X^*\|^2\right]$$
$$\geq D^* + W_2^2(p_{\hat{X}}, p_{X^*})$$
$$\geq D^* + (P^* - P)_+^2,$$

where the last inequality follows from (40). Hence $D(P) = \min_{p_{\hat{X}|Y}:W_2(p_{\hat{X}},p_X)\leq P} \mathbb{E}_{p_{\hat{X}Y}}\left[\|X-\hat{X}\|^2\right] \geq D^* + [(P^* - P)_+]^2$. □

### B.1.1 Proof of Theorem 3

**Theorem. 3**. *Let $\hat{X}_0$ be an estimator achieving perception index $0$ and MSE $D(0)$. Then for any $P \in [0, P^*]$, the estimator*

$$\hat{X}_P = \left(1 - \frac{P}{P^*}\right)\hat{X}_0 + \frac{P}{P^*}X^* \tag{41}$$

*is optimal for perception index $P$, namely, it achieves perception index $P$ and distortion $D(P)$.*

Let us prove a stronger result, from which Theorem 3 will follow.

**Lemma 5.** *Let $\hat{X}_\varepsilon$ be an estimator (independent of $X$ given $Y$) achieving $W_2(p_X, p_{\hat{X}_\varepsilon}) \leq \varepsilon_P$ and $\mathbb{E}\left[\|\hat{X}_\varepsilon - X^*\|^2\right] \leq (1 + \varepsilon_D)^2 W_2^2(p_X, p_{X^*})$ for some $\varepsilon_D, \varepsilon_P \geq 0$. Given $0 \leq P \leq P^* = W_2(p_X, p_{X^*})$, consider the estimator*

$$\hat{X}_P = \left(1 - \frac{P}{P^*}\right)\hat{X}_\varepsilon + \frac{P}{P^*}X^*. \tag{42}$$

*Then $\hat{X}_P$ achieves $\mathbb{E}[\|X - \hat{X}_P\|^2] \leq D^* + (1 + \varepsilon_D)^2(P^* - P)^2$ with perception index $\varepsilon_P + (1 + \varepsilon_D)P$. When $\varepsilon_D, \varepsilon_P = 0$, namely $\hat{X}_\varepsilon$ is an optimal perfect perceptual quality estimator, $\hat{X}_P$ is an optimal estimator under perception constraint $P$, which proves Theorem 3.*

*Proof.* $W_2^2(p_{\hat{X}_\varepsilon}, p_{\hat{X}_P}) \leq \mathbb{E}\left[\|\hat{X}_\varepsilon - \hat{X}_P\|^2\right]$, and using the triangle inequality

$$W_2(p_X, p_{\hat{X}_P}) \leq W_2(p_X, p_{\hat{X}_\varepsilon}) + W_2(p_{\hat{X}_\varepsilon}, p_{\hat{X}_P})$$
$$\leq \varepsilon_P + \sqrt{\mathbb{E}\left[\|\hat{X}_\varepsilon - \hat{X}_P\|^2\right]}$$
$$= \varepsilon_P + \sqrt{\frac{P^2}{W_2^2(p_X, p_{X^*})}\mathbb{E}\left[\|\hat{X}_\varepsilon - X^*\|^2\right]}$$
$$\leq \varepsilon_P + P(1 + \varepsilon_D),$$

where the equality is based on (42). A direct calculation of the distortion yields

$$\mathbb{E}\left[\|X^* - \hat{X}_P\|^2\right] = \left(1 - \frac{P}{W_2(p_X, p_{X^*})}\right)^2 \mathbb{E}\left[\|X^* - \hat{X}_\varepsilon\|^2\right]$$

$$\leq (1 + \varepsilon_D)^2 (W_2(p_X, p_{X^*}) - P)^2,$$

$$\mathbb{E}\left[\|X - \hat{X}_P\|^2\right] = D^* + \mathbb{E}\left[\|X^* - \hat{X}_P\|^2\right]$$

$$\leq D^* + (1 + \varepsilon_D)^2 (W_2(p_X, p_{X^*}) - P)^2.$$

When $\varepsilon_D, \varepsilon_P = 0$ we have $W_2(p_X, p_{\hat{X}_P}) \leq P$ and $\mathbb{E}\left[\|X - \hat{X}_P\|^2\right] \leq D^* + (W_2(p_X, p_{X^*}) - P)^2$. From Lemma 4, the latter inequality is achieved with equality. Note that since here $\mathbb{E}\left[\|\hat{X}_\varepsilon - X^*\|^2\right] = W_2^2(p_X, p_{X^*})$, the distributions of $\{\hat{X}_P, \ P \in [0, W_2(p_X, p_{X^*})]\}$ form a constant-speed geodesic, hence $W_2(p_X, p_{\hat{X}_P}) = P$. $\qquad\square$

**Corollary 1.** *When $X^*$ has a density, $\hat{X}_0$ (hence $\hat{X}_P$) can be obtained via a deterministic transformation of $Y$.*

*Proof.* Since the distribution of $X^*$ is absolutely continuous, we have an optimal map $T_{p_{X^*} \to p_X}$ between the distributions of $X^*$ and $X$ (see discussion in App. A.2). Namely, we have that $\hat{X}_0 = T_{p_{X^*} \to p_X}(X^*)$ is an optimal estimator with perception index 0. Thus, according to (15) $\hat{X}_P = \left(1 - \frac{P}{P^*}\right) T_{p_{X^*} \to p_X}(X^*) + \frac{P}{P^*} X^*$ are optimal estimators, which in this case are given by a deterministic function of $Y$. $\qquad\square$

## B.2 Proof of Theorem 1

With Theorem 3 and Lemma 5 in hand, we are now ready to prove our main result.

**Theorem. 1**. *The DP function (2) is given by*

$$D(P) = D^* + [(P^* - P)_+]^2. \tag{43}$$

*Furthermore, an estimator achieving perception index $P$ and distortion $D(P)$ can always be constructed by applying a (possibly stochastic) transformation to $X^*$.*

*Proof.* When $P \geq P^*$ the result is trivial since $D(P) = D^*$. Let us focus on $P < P^*$. Since $X, X^* \in \mathbb{R}^{n_x}$, we have an optimal plan $p_{\hat{X}_0 X^*}$ between their distributions, attaining $P^*$ [2, 20]. We then have an optimal estimator $\hat{X}_0$ with perception index 0, which is given by this joint distribution hence achieving $\mathbb{E}\left[\|\hat{X}_0 - X^*\|^2\right] = (P^*)^2$ (for the connection between $p_{\hat{X}_0 X^*}$ and the choice of $p_{\hat{X}_0|Y}$, see Remark about uniqueness in Sec. 3.1). For any perception $P < P^*$, consider $\hat{X}_P$ given by (41). We have $W_2(p_X, p_{\hat{X}_P}) = P$, and (see Theorem 3's proof)

$$\mathbb{E}\left[\|X - \hat{X}_P\|^2\right] \leq D^* + (W_2(p_X, p_{X^*}) - P)^2,$$

hence $D(P) \leq D^* + [(P^* - P)_+]^2$. On the other hand, we have (Lemma 4) $D(P) \geq D^* + [(P^* - P)_+]^2$, which completes the proof. $\qquad\square$

## B.3 The Gaussian setting

In this Section we prove Theorems 4 and 5. We begin by proving Theorem 5, and then show that Theorem 4 follows as a special case. Recall that

$$(G^*)^2 = \text{Tr}\left\{\Sigma_X + \Sigma_{X^*} - 2\left(\Sigma_X^{1/2}\Sigma_{X^*}\Sigma_X^{1/2}\right)^{1/2}\right\} \tag{44}$$

and

$$T^* = \Sigma_X^{-1/2}(\Sigma_X^{1/2}\Sigma_{X^*}\Sigma_X^{1/2})^{1/2}\Sigma_X^{-1/2}. \tag{45}$$

**Theorem. 5.** *Consider the setting of Theorem 4 in the main text. Let* $\Sigma_{\hat{X}_0 Y} \in \mathbb{R}^{n_x \times n_y}$ *satisfy*

$$\Sigma_{\hat{X}_0 Y} \Sigma_Y^{-1} \Sigma_{YX} = \Sigma_X^{\frac{1}{2}} (\Sigma_X^{\frac{1}{2}} \Sigma_{X^*} \Sigma_X^{\frac{1}{2}})^{\frac{1}{2}} \Sigma_X^{-\frac{1}{2}}, \tag{46}$$

*and* $W_0$ *be a zero-mean Gaussian noise with covariance*

$$\Sigma_{W_0} = \Sigma_X - \Sigma_{\hat{X}_0 Y} \Sigma_Y^{-1} \Sigma_{\hat{X}_0 Y}^T \succeq 0 \tag{47}$$

*that is independent of* $Y, X$*. Then, for any* $P \in [0, G^*]$*, an optimal estimator with perception index* $P$ *can be obtained by*

$$\hat{X}_P = \left( \left(1 - \frac{P}{G^*}\right) \Sigma_{\hat{X}_0 Y} + \frac{P}{G^*} \Sigma_{XY} \right) \Sigma_Y^{-1} Y + \left(1 - \frac{P}{G^*}\right) W_0. \tag{48}$$

*The estimator given in (50) is one solution to (46)-(47), but it is generally not unique.*

*Proof.* (Theorem 5) Let $\hat{X}_0 \triangleq \Sigma_{\hat{X}_0 Y} \Sigma_Y^{-1} Y + W_0$ where $\Sigma_{\hat{X}_0 Y}$ satisfies (46)-(47). It is easy to see that $\hat{X}_0 \sim \mathcal{N}(0, \Sigma_X)$ and it is jointly Gaussian with $(X, Y, X^*)$. We have by (46)

$$\mathbb{E}\left[X^* \hat{X}_0^T\right] = \Sigma_{XY} \Sigma_Y^{-1} \Sigma_{Y \hat{X}_0} = \Sigma_X^{-1/2} (\Sigma_X^{1/2} \Sigma_{X^*} \Sigma_X^{1/2})^{1/2} \Sigma_X^{1/2}, \tag{49}$$

hence using (47),

$$\begin{aligned}
\mathbb{E}\left[\|\hat{X}_0 - X^*\|^2\right] &= \mathrm{Tr}\left\{\Sigma_X + \Sigma_{X^*} - 2\mathbb{E}\left[X^* \hat{X}_0^T\right]\right\} \\
&= \mathrm{Tr}\left\{\Sigma_X + \Sigma_{X^*} - 2\Sigma_X^{-1/2}(\Sigma_X^{1/2} \Sigma_{X^*} \Sigma_X^{1/2})^{1/2} \Sigma_X^{1/2}\right\} \\
&= \mathrm{Tr}\left\{\Sigma_X + \Sigma_{X^*} - 2(\Sigma_X^{1/2} \Sigma_{X^*} \Sigma_X^{1/2})^{1/2}\right\} \\
&= G^2(\Sigma_X, \Sigma_{X^*}) \\
&= (G^*)^2.
\end{aligned}$$

Summarizing, $\hat{X}_0$ is an optimal perfect perceptual quality estimator. Note that (48) can be written as

$$\hat{X}_P = \left(1 - \frac{P}{G^*}\right) \hat{X}_0 + \frac{P}{G^*} X^*,$$

and by Theorem 3 we have that it is an optimal estimator. $\qquad\square$

Before proceeding to the proof of Theorem 4, let us introduce some auxiliary facts.

**Lemma 6.** *Let* $\Sigma, \Sigma_{X^*} \in \mathbb{R}^{n \times n}$ *be (symmetric) PSD matrices, and* $\Sigma_X \in \mathbb{R}^{n \times n}$ *is PD. Denote* $T^* = \Sigma_X^{-\frac{1}{2}}\left(\Sigma_X^{\frac{1}{2}} \Sigma_{X^*} \Sigma_X^{\frac{1}{2}}\right)^{\frac{1}{2}} \Sigma_X^{-\frac{1}{2}}$*. Then:*

1. $\mathrm{Ker}\{\Sigma\} = \mathrm{Ker}\{\Sigma^{\frac{1}{2}}\}$.

2. $\mathrm{Ker}\{\Sigma_*\} \subseteq \mathrm{Ker}\{\Sigma_X^{\frac{1}{2}}(\Sigma_X^{\frac{1}{2}} \Sigma_{X^*} \Sigma_X^{\frac{1}{2}})^{\frac{1}{2}} \Sigma_X^{-\frac{1}{2}}\} = \mathrm{Ker}\{\Sigma_X T^*\}$, *and we have* $\Sigma_X T^* \Sigma_{X^*}^\dagger \Sigma_{X^*} = \Sigma_X T^*$.

*Proof.* (1) Let $\Sigma$ be PSD. Since it is real and symmetric it is diagonalizable, $\Sigma = UDU^T$ and $\Sigma^{1/2} = UD^{1/2}U^T$ where $D$ is a diagonal matrix with non-negative entries which are the eigenvalues of $\Sigma$. We have $\mathrm{Ker}\{D\} = \mathrm{Ker}\{D^{1/2}\} = \{v \in \mathbb{R}^n : v_i = 0 \,\forall i : D_{i,i} \neq 0\}$ and since $U$ is full-rank, $\mathrm{Ker}\{\Sigma\} = \mathrm{Ker}\{\Sigma^{1/2}\} = U\mathrm{Ker}\{D\}$.

(2) Assume $\Sigma_{X^*} v = 0$. We have $(\Sigma_X^{1/2} \Sigma_{X^*} \Sigma_X^{1/2}) \Sigma_X^{-1/2} v = 0$, implying that $\Sigma_X^{-1/2} v \in \mathrm{Ker}\{(\Sigma_X^{1/2} \Sigma_{X^*} \Sigma_X^{1/2})\} = \mathrm{Ker}\{(\Sigma_X^{1/2} \Sigma_{X^*} \Sigma_X^{1/2})^{1/2}\}$. The equality is true since $\Sigma_X^{1/2} \Sigma_{X^*} \Sigma_X^{1/2} = \Sigma_X^{1/2} \Sigma_{X^*}^{1/2} (\Sigma_X^{1/2} \Sigma_{X^*}^{1/2})^T$ is PSD, and we use (1). To conclude, we have

$$\Sigma_X T^* v = \Sigma_X^{1/2} (\Sigma_X^{1/2} \Sigma_{X^*} \Sigma_X^{1/2})^{1/2} \Sigma_X^{-1/2} v = 0 \implies \mathrm{Ker}\{\Sigma_{X^*}\} \subseteq \mathrm{Ker}\{\Sigma_X T^*\}.$$

Recall now that $(I - \Sigma_{X^*}^\dagger \Sigma_{X^*})$ is a projection onto $\mathrm{Ker}\{\Sigma_{X^*}\}$. We have $\Sigma_X T^*(I - \Sigma_{X^*}^\dagger \Sigma_{X^*}) = 0$, yielding $\Sigma_X T^* \Sigma_{X^*}^\dagger \Sigma_{X^*} = \Sigma_X T^*$. $\qquad\square$

The following Lemma is a reminder of the Schur Complement and its properties.

**Lemma 7.** *[Schur complement]. Let* $\Sigma = \begin{bmatrix} A & B \\ B^T & C \end{bmatrix}$ *be a symmetric matrix where $A$ is PD. Then* $\Sigma/A \triangleq C - B^T A^{-1} B$ *is the Schur complement of $\Sigma$, and we have that $\Sigma$ is PSD iff $\Sigma/A$ is PSD.*

We are now ready to prove Theorem 4.

**Theorem. 4.** *Assume $X$ and $Y$ are zero-mean jointly Gaussian random vectors with $\Sigma_X, \Sigma_Y \succ 0$. Then for any $P \in [0, G^*]$, an estimator with perception index $P$ and MSE $D(P)$ can be constructed as*

$$\hat{X}_P = \left( \left(1 - \frac{P}{G^*}\right) \Sigma_X^{\frac{1}{2}} \left(\Sigma_X^{\frac{1}{2}} \Sigma_{X^*} \Sigma_X^{\frac{1}{2}}\right)^{\frac{1}{2}} \Sigma_X^{-\frac{1}{2}} \Sigma_{X^*}^{\dagger} + \frac{P}{G^*} I \right) \Sigma_{XY} \Sigma_Y^{-1} Y + \left(1 - \frac{P}{G^*}\right) W, \quad (50)$$

*where $W$ is a zero-mean Gaussian noise with covariance $\Sigma_W = \Sigma_X^{1/2}(I - \Sigma_X^{1/2} T^* \Sigma_{X^*}^{\dagger} T^* \Sigma_X^{1/2}) \Sigma_X^{1/2}$, which is independent of $Y, X$.*

*Proof.* We observe that (50) is a special case of (48), where $\Sigma_{\hat{X}_0 Y} = \Sigma_{Y \hat{X}_0}^T = \Sigma_X^{\frac{1}{2}} \left(\Sigma_X^{\frac{1}{2}} \Sigma_{X^*} \Sigma_X^{\frac{1}{2}}\right)^{\frac{1}{2}} \Sigma_X^{-\frac{1}{2}} \Sigma_{X^*}^{\dagger} \Sigma_{XY}$. We now show that $\Sigma_{\hat{X}_0 Y}$ has the desired properties (46)-(47). By substitution,

$$\Sigma_{\hat{X}_0 Y} \Sigma_Y^{-1} \Sigma_{YX} = \Sigma_X^{\frac{1}{2}} \left(\Sigma_X^{\frac{1}{2}} \Sigma_{X^*} \Sigma_X^{\frac{1}{2}}\right)^{\frac{1}{2}} \Sigma_X^{-\frac{1}{2}} \Sigma_{X^*}^{\dagger} \left(\Sigma_{XY} \Sigma_Y^{-1} \Sigma_{YX}\right)$$

$$= \Sigma_X^{\frac{1}{2}} \left(\Sigma_X^{\frac{1}{2}} \Sigma_{X^*} \Sigma_X^{\frac{1}{2}}\right)^{\frac{1}{2}} \Sigma_X^{-\frac{1}{2}} \Sigma_{X^*}^{\dagger} \Sigma_{X^*}$$

$$= \Sigma_X^{\frac{1}{2}} \left(\Sigma_X^{\frac{1}{2}} \Sigma_{X^*} \Sigma_X^{\frac{1}{2}}\right)^{\frac{1}{2}} \Sigma_X^{-\frac{1}{2}}.$$

The last equality is due to Lemma 6.

Recall $\Sigma_{X^*}^{\dagger} \Sigma_{X^*} \Sigma_{X^*}^{\dagger} = \Sigma_{X^*}^{\dagger}$, and we denote $T^* = \Sigma_X^{-\frac{1}{2}} \left(\Sigma_X^{\frac{1}{2}} \Sigma_{X^*} \Sigma_X^{\frac{1}{2}}\right)^{\frac{1}{2}} \Sigma_X^{-\frac{1}{2}}$. We now have

$$\Sigma_{Y\hat{X}_0} \Sigma_X^{-1} \Sigma_{\hat{X}_0 Y} = \Sigma_{YX} \Sigma_{X^*}^{\dagger} T^* \Sigma_X \Sigma_X^{-1} \Sigma_X T^* \Sigma_{X^*}^{\dagger} \Sigma_{XY}$$

$$= \Sigma_{YX} \Sigma_{X^*}^{\dagger} \Sigma_X^{-\frac{1}{2}} (\Sigma_X^{\frac{1}{2}} \Sigma_{X^*} \Sigma_X^{\frac{1}{2}}) \Sigma_X^{-\frac{1}{2}} \Sigma_{X^*}^{\dagger} \Sigma_{XY}$$

$$= \Sigma_{YX} \Sigma_{X^*}^{\dagger} \Sigma_{X^*} \Sigma_{X^*}^{\dagger} \Sigma_{XY}$$

$$= \Sigma_{YX} \Sigma_{X^*}^{\dagger} \Sigma_{XY},$$

hence

$$\Sigma_Y - \Sigma_{Y\hat{X}_0} \Sigma_X^{-1} \Sigma_{\hat{X}_0 Y} = \Sigma_Y - \Sigma_{YX} \Sigma_{X^*}^{\dagger} \Sigma_{XY} = \Sigma_{Y|X^*} \succeq 0. \quad (51)$$

Since $\Sigma_X, \Sigma_Y \succ 0$, (51) is the Schur complement of $\begin{bmatrix} \Sigma_X & \Sigma_{\hat{X}_0 Y} \\ \Sigma_{Y\hat{X}_0} & \Sigma_Y \end{bmatrix} \succeq 0$, yielding

$$\Sigma_W = \Sigma_X - \Sigma_{\hat{X}_0 Y} \Sigma_Y^{-1} \Sigma_{\hat{X}_0 Y}^T \succeq 0. \quad (52)$$

$\square$

**Corollary 2** (Non-singular special case). *In the case where $\Sigma_{X^*}$ is invertible, $\Sigma_{\hat{X}_0 Y} = \Sigma_X T^* \Sigma_{X^*}^{-1} \Sigma_{XY}$ in the proof of Theorem 4, and it is easy to see that the noise covariance is $\Sigma_W = 0$. In this case $\Sigma_{\hat{X}_0 Y}$ is the unique solution to (46)-(47). This means that $\hat{X}_0$ (hence $\hat{X}_P$) is a deterministic function of $Y$.*

*Proof.* We first show $\Sigma_W = 0$. Let $M_P = \Sigma_{\hat{X}_0 Y} = \Sigma_X T^* \Sigma_{X^*}^{-1} \Sigma_{XY}$, then

$$\Sigma_W = \Sigma_X - M_P \Sigma_Y^{-1} M_P^T$$

$$= \Sigma_X - \Sigma_X T^* \Sigma_{X^*}^{-1} \Sigma_{XY} \Sigma_Y^{-1} \Sigma_{YX} \Sigma_{X^*}^{-1} T^* \Sigma_X$$

$$= \Sigma_X - \Sigma_X \Sigma_X^{-1/2} (\Sigma_X^{1/2} \Sigma_{X^*} \Sigma_X^{1/2})^{1/2} \underbrace{(\Sigma_X^{-1/2} \Sigma_{X^*}^{-1} \Sigma_X^{-1/2})}_{=(\Sigma_X^{1/2} \Sigma_{X^*} \Sigma_X^{1/2})^{-1}} (\Sigma_X^{1/2} \Sigma_{X^*} \Sigma_X^{1/2})^{1/2} \Sigma_X^{-1/2} \Sigma_X$$

$$= \Sigma_X - \Sigma_X \Sigma_X^{-1/2} \Sigma_X^{-1/2} \Sigma_X = 0.$$

Now, assume $M$ is a solution to (46)-(47), then $M_\Delta = M_P - M$ satisfies $M_\Delta \Sigma_Y^{-1} \Sigma_{YX} = 0$ and

$$\Sigma_X - M\Sigma_Y^{-1}M^T =$$
$$\Sigma_X - [M_P\Sigma_Y^{-1}M_P^T + M_\Delta\Sigma_Y^{-1}M_\Delta^T - M_\Delta\Sigma_Y^{-1}M_P^T - M_P\Sigma_Y^{-1}M_\Delta^T] \succeq 0.$$

But, $M_\Delta\Sigma_Y^{-1}M_P^T = (M_\Delta\Sigma_Y^{-1}\Sigma_{YX})\Sigma_{X^*}^{-1}T^*\Sigma_X = 0$ and $\Sigma_X - M_P\Sigma_Y^{-1}M_P^T = 0$, yielding $M_\Delta\Sigma_Y^{-1}M_\Delta^T \preceq 0$. Since $M_\Delta\Sigma_Y^{-1}M_\Delta^T$ is PSD and $\Sigma_Y^{-1}$ is PD, we conclude that $M_\Delta = 0$. $\qquad\square$

## C   Relations with other divergences

While in Section 3 we focused our attention on the $MSE - W_2$ tradeoff, in this section we discuss the implications of our results on the DP tradeoff with other divergences. In particular, we show that when considering the MSE distortion, (8) establishes a lower bound on a class of DP functions. Note that at the point $P = 0$, the DP function coincides with (8) for all plausible divergences.

Let $d_p(\cdot, \cdot)$ be a divergence between probability measures, and let $D_{d_p}(P)$ be the DP function w.r.t. this divergence, given by (1), where MSE is used to measure distortion. Here, $D(P)$ will denote $D_{W_2}(P)$, given by (8). We can now write, similarly to (14),

$$D_{d_p}(P) = D^* + \inf_{d_p(p_X, p_{\hat{X}}) \leq P} W_2^2(p_{\hat{X}}, p_{X^*}). \tag{53}$$

In cases where $d_p(p_X, p_{\hat{X}}) \geq W_2(p_X, p_{\hat{X}})$ for all $p_{\hat{X}}$, the constraint set $\{p_{\hat{X}} : d_p(p_X, p_{\hat{X}}) \leq P\}$ is contained in $\{p_{\hat{X}} : W_2(p_X, p_{\hat{X}}) \leq P\}$. Therefore, from (53), we have that

$$D_{d_p}(P) \geq D^* + \inf_{W_2(p_X, p_{\hat{X}}) \leq P} W_2^2(p_{\hat{X}}, p_{X^*}) = D(P). \tag{54}$$

The last equality follows from (14), where the infimum is attained. The above result holds true for any Wasserstein distance $W_p$ with $p \geq 2$, since when $p \geq q \geq 1$, we have that $W_p(p_X, p_{\hat{X}}) \geq W_q(p_X, p_{\hat{X}})$ for all $p_{\hat{X}}, p_X$ [20].

For the case of $W_1$, let us denote $P_1^* \triangleq W_1(p_X, p_{X^*})$. From the triangle inequality, for every estimator satisfying $W_1(p_X, p_{\hat{X}}) \leq P$ we have

$$P_1^* \leq W_1(p_X, p_{\hat{X}}) + W_1(p_{\hat{X}}, p_{X^*}) \leq P + W_2(p_{\hat{X}}, p_{X^*}),$$

which together with (53) yields

$$D(P) \geq D_{W_1}(P) \geq D^* + [(P_1^* - P)_+]^2. \tag{55}$$

A similar result can be obtained for any $W_p$, $p \in [1, 2]$.

Note that when the support of $p_X$ and $p_{\hat{X}}$ is compact with diameter $R$, we have $R^{(p-q)/p}W_q^{q/p}(p_{\hat{X}}, p_X) \geq W_p(p_{\hat{X}}, p_X)$ for any $p \geq q \geq 1$ [20]. Particularly, $R^{1/2}W_1^{1/2}(p_{\hat{X}}, p_X) \geq W_2(p_{\hat{X}}, p_X)$, and therefore $W_1(p_{\hat{X}}, p_X) \leq P$ implies $W_2(p_{\hat{X}}, p_X) \leq \sqrt{RP}$, so we have from (53) that

$$D_{W_1}(P) \geq D(\sqrt{RP}). \tag{56}$$

In the Gaussian setting where $X \sim \mathcal{N}(0, I)$, we have by Talagrand's Inequality [28, 19] $W_2(p_{\hat{X}}, p_X) \leq \sqrt{2d_{KL}(p_{\hat{X}}\|p_X)}$ for $p_{\hat{X}} \ll p_X$, hence we obtain, similarly to (54)

$$D_{d_{KL}}(P) \geq D(\sqrt{2P}). \tag{57}$$

We summarize these results in Appendix F.

## D   Settings with commuting covariances

In many practical problems, covariance matrices may have the commutative relation $\Sigma_X \Sigma_{X^*} = \Sigma_{X^*} \Sigma_X$. This is the case, for example, of circulant or large Toeplitz matrices [9]. For natural images

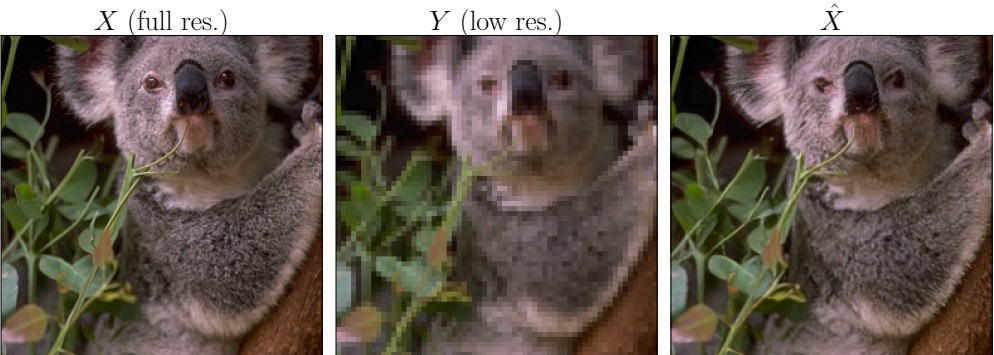

$X$ (full res.)  $Y$ (low res.)  $\hat{X}$

Figure 5: A visual demonstration of SR image enhancement. $X$ is a full-resolution reference image and $Y$ is a $\times 4$ downsampled version of $X$. $\hat{X}$ is a reconstruction of $X$ based on $Y$.

this is a reasonable assumption since shift-invariance induces diagonalization by the Fourier basis [30].

In the Gaussian settings of Sec. 3.3, where $\Sigma_X, \Sigma_{X^*}$ commute it is easy to see that the Gelbrich distance between them can be written as

$$G^* = G((\mu_X, \Sigma_X), (\mu_{X^*}, \Sigma_{X^*})) = \|\Sigma_X^{1/2} - \Sigma_{X^*}^{1/2}\|_F.$$

$\|A\|_F = \sqrt{\text{Tr}\{A^T A\}}$ is the Frobenius norm. This is due to the fact that $\Sigma_X^{1/2}, \Sigma_{X^*}^{1/2}$ also commute. In order to achieve $\mathbb{E}\left[\|\hat{X}_0 - X^*\|^2\right] = (G^*)^2$, an optimal perfect perceptual quality estimator has to satisfy (49) which now takes the form

$$\mathbb{E}\left[X^* \hat{X}_0^T\right] = \Sigma_X^{1/2} \Sigma_{X^*}^{1/2}.$$

It is easy to see that estimators obtained by $\hat{X}_0, X^*$ using (15) are Gaussian with zero mean and covariance $\Sigma_P$, given by

$$\Sigma_P^{\frac{1}{2}} = \left(1 - \frac{P}{G^*}\right) \Sigma_X^{\frac{1}{2}} + \frac{P}{G^*} \Sigma_{X^*}^{\frac{1}{2}}. \tag{58}$$

Pay attention that since the roots commute, $\Sigma_P$ commmutes with $\Sigma_X, \Sigma_{X^*}$, and

$$\|\Sigma_X^{\frac{1}{2}} - \Sigma_P^{\frac{1}{2}}\|_F = P, \quad \|\Sigma_P^{\frac{1}{2}} - \Sigma_{X^*}^{\frac{1}{2}}\|_F = G^* - P.$$

This further reduces the geometry of the problem to the $l^2$-distance between commuting matrices.

# E    Numerical illustration

## E.1    Super-resolution problem

In super-resolution (SR) problems, the objective is to enhance the resolution of a given image. This setting can be viewed as an image reconstruction problem, where we assume $X$ is an unknown image of the desired resolution, and the input to the algorithm is $Y$, a downsampled (degraded) version of $X$. The output of the algorithm is then $\hat{X} \sim p_{\hat{X}|Y}$, an estimation of $X$ based on $Y$.

Figure 5 visually demonstrates this setting with a concrete example.

## E.2 Simulation details

In Section 5 we construct an experimental setup, demonstrating our results. Figure 3 presents the evaluation of 13 super resolution algorithms on the BSD100 dataset, where we compare MSE distortion, and Gelbrich and FID perceptual indices. Low resolution images were obtained by $4\times$ downsampling BSD100 images using a bicubic kernel.

For each algorithm, we acquire 100 RGB images (5000 for the explorable SR method) which are reconstructions of BSD100 images. To compute the Gelbrich index, we extract $9 \times 9$ patches from the RGB images, and then estimate

$$m_{\text{Alg}} = \frac{1}{N_{\text{patches}}} \sum_i p_i, \quad \Sigma_{\text{Alg}} = \frac{1}{N_{\text{patches}} - 1} (p_i - m_{\text{Alg}})(p_i - m_{\text{Alg}})^T,$$

where $p_i$ is the $i$-th patch (a 243-row vector) and $N_{\text{patches}} = 1,643,200$. We compute using (4)

$$\text{MSE}_{\text{Alg}} = \frac{1}{243 \times N_{\text{patches}}} \sum_i \|p_i^{\text{Alg}} - p_i^{\text{BSD100}}\|^2, \quad P_{\text{Alg}} = \sqrt{\frac{1}{243}} G\left((m_{\text{BSD100}}, \Sigma_{\text{BSD100}}), (m_{\text{Alg}}, \Sigma_{\text{Alg}})\right).$$

The stochastic explorable SR method [3] is evaluated using 50 different SR outputs for each input image, hence for this method $N_{\text{patches}} = 50 \times 1,643,200$.

FID values are calculated on $299 \times 299$ patches, where for the explorable SR method we use 40 different outputs for each input.

The estimators $\hat{X}_t$ are constructed using per-pixel interpolation between EDSR and ESRGAN,

$$\hat{X}_t = tX_{\text{EDSR}} + (1 - t)X_{\text{ESRGAN}}.$$

## E.3 Visual illustration

Here we present a visual comparison between SR methods and our constructed estimators, achieving roughly the same MSE but with a lower perception index. We also present EDSR, ESRGAN, the low-resolution input, and the ground-truth BSD100 images.

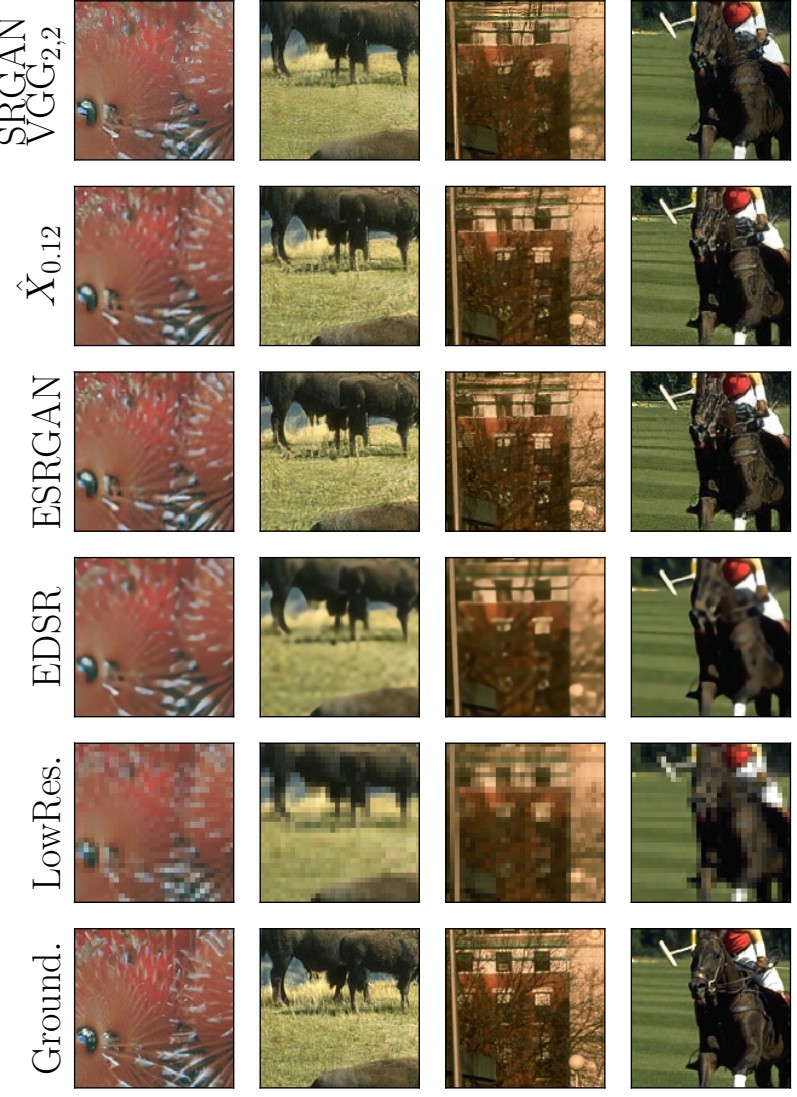

Figure 6: A visual comparison between SRGAN-VGG$_{2,2}$ (RMSE: 18.08, P: 5.05), and $\hat{X}_{0.12}$ (18.14, 2.59).

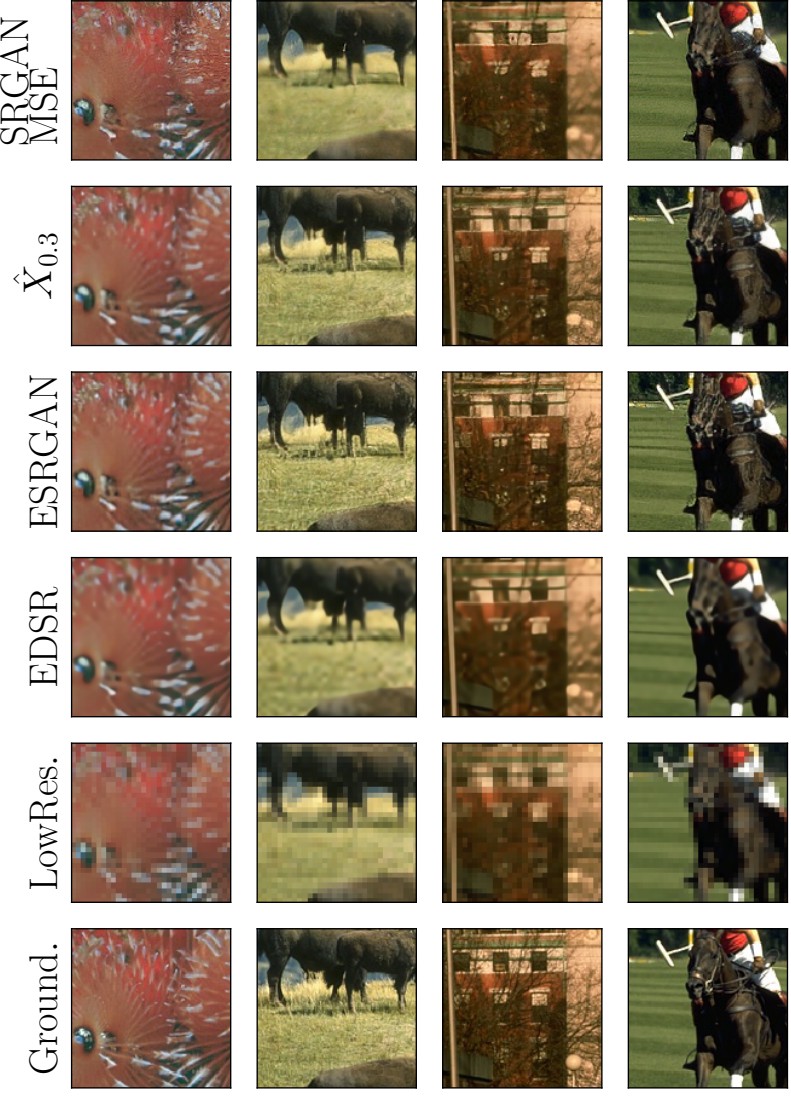

Figure 7: A visual comparison between SRGAN-MSE (RMSE: 16.93, P: 5.85), and $\hat{X}_{0.3}$ (16.82, 4.32).

# F   Table of main results

For convenience, we summarize our results in the following Table.

Table 1: Main results

| Result | notation | setting | | remarks |
|---|---|---|---|---|
| **D-P function** | $D(\mathrm{P})$ | MSE-$W_2$ | $D(P) = D^* + \left[(P^* - P)_+\right]^2$ | $P^* = W_2(p_X, p_{X^*})$ |
| | | Gaussian | $D(P) = D^* + \left[(G^* - P)_+\right]^2$ | $G^* = G(\Sigma_X, \Sigma_{X^*})$ |
| **Optimal estimators** | $\hat{X}_P$ | MSE-$W_2$ | $\left(1 - \frac{P}{P^*}\right)\hat{X}_0 + \frac{P}{P^*}X^*$ | |
| | | Gaussian | $\left(\alpha\Sigma_X T^*\Sigma_{X^*}^{\dagger} + (1-\alpha)I\right)X^*$ $+\alpha W$ | $\alpha = \left(1 - \frac{P}{G^*}\right), X^* = \Sigma_{XY}\Sigma_Y^{-1}Y$ $T^* = \Sigma_X^{-\frac{1}{2}}\left(\Sigma_X^{\frac{1}{2}}\Sigma_{X^*}\Sigma_X^{\frac{1}{2}}\right)^{\frac{1}{2}}\Sigma_X^{-\frac{1}{2}}$ $W \sim \mathcal{N}(0, \Sigma_X - \Sigma_X T^*\Sigma_{X^*}^{\dagger}T^*\Sigma_X)$ |
| **Lower bounds** | | MSE-$W_2$ | $D(P) \geq D^* + \left[(G^* - P)_+\right]^2$ | |
| | | MSE-$W_p$ | $D_{W_p}(P) \geq D^* + \left[(P^* - P)_+\right]^2$ | $p \geq 2$ |
| | | MSE-$W_1$ | $D_{W_1}(P) \geq D^* + \left[(P_1^* - P)_+\right]^2$ | $P_1^* = W_1(p_X, p_{X^*})$ |
| | | MSE-$d_{KL}$ | $D_{d_{KL}}(P) \geq D^* + \left[(P^* - \sqrt{2P})_+\right]^2$ | $X \sim \mathcal{N}(0, I)$ |