# OpenReview forum: "A Theory of the Distortion-Perception Tradeoff in Wasserstein Space"
_NeurIPS.cc/2021/Conference — NeurIPS 2021 Poster_

### Official Review · Reviewer_LmPX · 2021-07-16

**Rating:** 6
**Confidence:** 2

**Summary:**

The paper studies the trade-off between minimizing the mean-square error (distortion) and the
Wasserstein-2 distance (perception) when constructing an estimator \hat{X} of a random variable X.
In particular, the work analyzes a distortion-perception function, which given the radius P of a Wasserstein-2
ball around X returns the minimal mean-square error of \hat{X} constrained to be in that ball.

The analysis reveals that for estimators with small Wasserstein-2 distance to the true distribution, one can often
obtain an estimator with much better mean-square error by a simple linear interpolation in the output space.

The theoretical results are applied to deep image super-resolution, where it is shown that the perceptual quality of
an estimator with small MSE can be improved by interpolating it with one having small Wasserstein-2 distance. Remarkably,
the interpolated estimator often still has low MSE, while significantly lower Wasserstein-2 distance, leading to a greatly
improved perceptual quality.

**Limitations And Societal Impact:**

As mentioned in the main review, a discussion on the limitations of performing a simple linear interpolation in output space is missing.
Are there for example "failure cases", where perception and distortion measured by the Wasserstein-distance and MSE are good
but the visual quality is worse than of the two individual estimators?

I don't believe that the work needs to address negative impacts on society, as it is largely theoretical.


**Main Review:**

The paper is well-written, and I have no major objections regarding the overall clarity of the work.

One concern is that the approach is in some sense an incremental step compared to [4], which already introduces the perception-distortion tradeoff in image restoration. The main contribution over that work is the closed-form expression for the distortion-perception function for the specific combination of MSE and Wasserstein-2 distance.

To me, it appears that the significance / impact of the paper is overall rather limited, as the final estimator suggested by the theory is just a convex combination of the two estimator in output space. This seems to stem from the perhaps overly simplistic choice of MSE / Wasserstein-2, which ignores the fact that the manifold of natural images is a highly nonlinear space. Therefore, I feel that this strategy comes with obvious limitations and suggests that the considered measure of perception might not be too meaningful.

I apologize if the above criticisms are based on some basic misconception / incomplete understanding of the contributions and I am willing to raise my score in such a case.

Finally, I found the experimental evaluation a bit short. Qualitative results are only shown on four different images. For Figure 3, I do not agree that there is an improvement in perceptual quality as claimed, especially for the first image. Such claims could be more rigorously evaluated using a small-scale study involving humans.

### Minor comments.
- Due to the concise choice of notation I had difficulties following the mathematical
details starting from Section 2.1. There, X, \hat{X} and Y are introduced as random vectors, which mathematically are
measurable functions mapping from a space which being left unspecified into R^{n_x} and R^{n_y}. These introduce the pushforward measures
p_X, p_{\hat{X}} and a certain conditional distribution p_{\hat X | Y}. Since these objects are introduced in a quite implicit/indirect way,
mentioning the exact signature and definition of the density functions (or Borel probability measures?) would be helpful.

- It might be beneficial for the reader also to explain what X, \hat X and Y specialize to in the setting of a concrete example,
perhaps for the case of image super-resolution considered in the applications sections. Maybe such an example can be included after line 76 in the paper, or in the appendix.

- I felt a bit uneasy about the precise definition of Eq. (1). In particular, it is unclear with respect to which probability distribution
the expectation is taken (Is it wrt the unspecified distribution that X and \hat X pushforward or is it the conditional distribution?). What space does p_{\hat X | Y} live in?
Perhaps these things could be expanded on in a bit more detail in Appendix A.1?

- I can understand why an economical/concise notation is chosen (as is often in machine learning papers). But for me, it made checking the details in a rigorous fashion a bit exhausting. Overall I don't have doubts about the correctness of the results though.

- The distortion-perception trade-off is studied on the example of image restoration. This is a fascinating and important field,  but perhaps a slightly niche topic for the NeurIPS audience, limiting the overall significance and impact of the work. Could the proposed framework perhaps
considered in a broader context? For example, empirical risk minimization problems in machine learning optimize for low distortion; could generalization capabilities of ML methods perhaps be improved by a similar interpolation technique?

- l. 154. I did not understand what a deterministic function is.


**Time Spent Reviewing:**

10 hours

---

> ### Author Response · Authors · 2021-08-09
> **Response to Reviewer LmPX**
>
> Thank you for your comments.
>
> **Results are incremental compared to [4]:**
>
> Note that despite extensive recent research on the topic, our results were never obtained previously. We emphasize that in [4], no closed-form D(P) function or estimators were obtained, not even for the 1-D Gaussian case.
>
> In a related historical context, we note that rate-distortion theory was introduced by Shannon as early as 1948. But even today, only limited setups in this domain admit closed form analytic results, and it is still a very active field of research. In analogy, [4] introduced the perception-distortion problem, but has left many open questions. Some of these were addressed in the current work, and many others remain.
>
>
>
> **Choice of MSE / Wasserstein-2 is overly simple, the significance / impact of the convex combination is limited:**
>
> First, we emphasize that while the convex combination result may arise from the nature of the distortion and perception indices, it was not at all clear from the outset that this result is indeed true. Note that some of the reviewers pointed this result to be extremely powerful / useful.
>
> Second, the choice of Wasserstein-2 is not as limiting as it may seem, since  we can use our results to bound the D-P function w.r.t other divergences. In particular, (8) yields lower bounds on D(P) w.r.t. $W_p$ (for $p>2$), $W_1$ (with slight modifications), KL (when $p_X$ is Gaussian). Needless to say, at the point P=0 , D(P=0) coincides with (8) for all divergences.  We will add this discussion about other divergences (with MSE distortion) to the text.
>
> **The choice of MSE / Wasserstein-2, ignores the fact that the manifold of natural images is nonlinear:**
>
> First, please note that the estimators we obtain are nonlinear functions of the measurements $Y$. In particular, if $X$ lies on a low-dimensional manifold, then at perception P=0, we also obtain that $\hat{X}$ lies on the same manifold.
>
> Additionally, the Wasserstein-2 distance can be naturally extended to Riemannian manifolds (under the manifold metric), where Euclidean spaces are a special case (see e.g. [R1] below). We will discuss this possible future extension in the revised document.
>
> [R1] F. Otto, C. Villani, “Generalization of an Inequality by Talagrand and Links with the Logarithmic Sobolev Inequality”, 1999.
>
> **Experimental evaluation is short, I do not agree that there is an improvement in perceptual quality as claimed:**
>
> Please note that our reconstructions tend to be visually sharper, while achieving the same level of distortion. To further support our claim, we computed FID and Wasserstein-1 scores for the SR methods of Section 5. The FID score was explicitly designed to assess the visual quality of generated images [R2]. We created figures similar to Fig. 2 using these scores as perception indices, and the empirical results suggest again that our interpolated estimators are closer to the D-P curve under these divergences. Namely, our estimators possess a better perception index with a similar MSE for both divergences. We will add these figures to the Appendix.
>
> [R2] Heusel et al. "GANs Trained by a Two Time-Scale Update Rule Converge to a Local Nash Equilibrium", Advances in neural information processing systems 30 (2017).
>
> **Minor comments:**
>
> **Concise choice of notation:**
> Note that the paper is written for both a theoretically oriented audience and a more practical one. As such, it is not always easy to find the appropriate balance between rigor and intuition. We chose to use the common notation in the estimation literature. But we will clarify the precise definitions for all quantities we use.
>
> **Explaining what $X, \hat X$ and $Y$ are on a concrete example:**
> In the setting of SR tasks, for example, $X$ is a high-resolution image we have no access to. $Y$ is a downscaled (smaller) version of $X$, which is what we measure. $\hat{X}$ is a restored image (the output of a restoration algorithm that accepts $Y$ as input). We will add a detailed explanation of the problem setting for the broader audience in the Appendix. We Will also illustrate the SR scenario (what are $X,Y,\hat X$ etc.) in a more visual manner using a concrete example.
>
>
> **Precise definition of Eq. (1):**
>
> Thank you for this comment, we will clarify the definition of (1) in the text.
>
> First, note that we use the convention that all expectations are w.r.t. the joint distribution of all random variables within the square brackets. We’ll clarify this.
>
> Specifically in the context of (1), the expectation is taken w.r.t. the joint probability $p_{X,\hat X}$, induced by $p_{\hat{X }|Y}$,  $p_{X |Y}$, and $p_Y$ assuming that $\hat X$ and $X$ are independent given $Y$. Thus, for example, if all distributions have densities, then $p_{X,\hat X}(x,\hat x)=\int p_{\hat{X} |Y}(\hat x|y) p_{X |Y}(x|y) p_Y(y) dy $. In the general case, the measure $p_{X,\hat X}$ is also defined, through the disintegration theorem [2].
>
> **Could the proposed framework perhaps be considered in a broader context, like ERM?**
> Note that our work assumes known distributions, while ERM methods rely on unknown distributions and on statistical guarantees based on finite data samples. But this is certainly an intriguing topic for future research.
>
> **What a deterministic function is in l. 154:**
> By a ‘deterministic function’ we mean that $p_{\hat X|Y=y} = \delta_{\hat X = f(y)}$ for some measurable function f: $R^{n_y}\rightarrow R^{n_x}$, where $\delta$ is the atomic measure (a.k.a Dirac’s delta distribution). We acknowledge that in common mathematical jargon, a ‘function’ is always considered to be deterministic.
>
> **Are there for example "failure cases", where perception and distortion measured by the Wasserstein-distance and MSE are good but the visual quality is worse than of the two individual estimators?**
>
> First, as we prove in Theorem 1, the MSE and the Wasserstein distance cannot both be small, as there is a tradeoff between them. Second, it is impossible to obtain better visual quality than the estimator $\hat{X_0}$ (corresponding to $P=0$) because it has the exact same distribution as $X$. Namely, an observer cannot tell apart images drawn from $p_X$ and images drawn from $p_{\hat{X}_0}$. As for obtaining worse visual quality than $X^*$, this might indeed be possible as the Wasserstein distance (for $P>0$) might not always fully correlate with human perception. We will discuss this limitation.

---

### Official Review · Reviewer_WGqG · 2021-07-16

**Rating:** 7
**Confidence:** 3

**Summary:**

The paper characterizes the perception-distortion in the context of (image) enhancement tradeoff, complementing prior results. For example, in image super resolution, this tradeoff describes the relationship between the point-wise reconstruction quality (distortion) and how well the distribution of the reconstructions match the distribution of the high-resolution data one aims to reconstruct. The paper derives this general tradeoff explicitly and characterizes certain aspects specifically for Gaussian distributions. The theoretical results are illustrated with experiments.

**Limitations And Societal Impact:**

For limitations please see the main review. In terms of societal impact it might make sense to briefly mention the potential issues inherent with perceptual super-resolution as possibly unwanted image content might be synthesized. In that context, the present paper is an advancement as it derives a method to adjust the D(P) tradeoff.

**Main Review:**

To my knowledge, the paper presents novel results characterizing the perception-distortion tradeoff for image (or more generally signal) enhancement as introduced by [4]. The paper focuses on MSE as a distortion and the Wasserstein-2 distance to measure perceptual quality. I particularly like that the authors derive Theorem 3 which provides practical guidance as to how to navigate the distortion-perception tradeoff. The paper is well-written.

I have the following questions and suggestions:
- MSE and the Wasserstein-2 distance are very specific choices of rate and distortion, and most super-resolution models are trained with different distortion metrics and/or GAN losses approximating a different divergence. What can be said about these cases? Any intuition whether similar properties as those proven here could be expected? This could also be empirically evaluated in the context of the experiments described in Sec 5. For example the authors could in addition use the Frechet Inception Distance as an additional perceptual index.
- The experiments in Sec 5 are done for deterministic decoders. How do the practical implications of Thm 3 (15) change for stochastic decoders as in [15, 3, 21]?
- It is sometimes unclear whether the m_i and \Sigma_i just refer to the first and second moments of a distribution (that could e.g. model natural images) or whether specifically Gaussian distributions are meant. This is in particular important because Guassians are a crude model for natural images (in the context of super resolution).
- In Sec 3.1 the authors argue that D^\star can be estimated using the loss of an MSE trained network after training. However, this value is usually an upper bound on D^\star and plugging it into (9) will invalidate the inequality. This should be pointed out clearly in the paper. If possible, it would be great to make a corresponding practical statement where one fixes the predictor and then derives a lower bound on D(P) for that predictor.
- I feel that the first statement in the abstract sounds too general. It should be mentioned that it is in the context of image restoration (or at least a problem that admits a distortion-perception tradeoff)

Overall I think the paper makes an interesting contribution.


**Time Spent Reviewing:**

4

---

> ### Author Response · Authors · 2021-08-09
> **Response to Reviewer WGqG**
>
> Thank you for your comments.
>
> **Does the D-P function for other measures behave similarly to the $MSE - W_2$ case?**
>
> Thanks, this is a very interesting point.
>
> We currently do not have a proof for an interpolation property for other distortions and divergences.
> However, please note that the experiments of Section 5 were performed with pre-trained SR models that were not trained with $MSE + W_2$ loss, yet the interpolation still yields visually good reconstructions. In any case, we’ll add more empirical and theoretical discussions of other divergences.
>
> Empirically, we evaluated FID and $W_1$ scores for the SR methods of Section 5. We created figures similar to Fig. 2 using these scores as perception indices. The results suggest that, again, interpolated estimators are close to the D-P curve under these divergences. We will add these figures to the Appendix.
>
> Theoretically, the choice of Wasserstein-2 is not as limiting as it may seem, since  we can use our results to bound the D-P function w.r.t other divergences. In particular, (8) yields lower bounds on D(P) w.r.t. $W_p$ (for $p>2$), $W_1$ (with slight modifications), KL (when $p_X$ is Gaussian). Needless to say, at the point P=0 , D(P=0) coincides with (8) for all divergences.  We will add this discussion about other divergences (with MSE distortion) to the text.
>
> **Stochastic estimators:** When we take the decoder approximating $\hat{X}_0$ to be stochastic, the results are qualitatively unchanged. We’ll add an illustration of this in Section 5.
>
> **Clarification of notations $m_i$ and $\Sigma_i$:**
> $m_i$ and $\Sigma_i$ refer to the first and second moments of a distribution. Wherever distributions are further assumed to be Gaussian, we explicitly mention it. We will try to clarify this (e.g. in l.136, l.169, l.284).
>
> **Approximation of lower bound (9):**
> You are correct. Note that we do mention that (9) only approximates a lower bound (l.139). But we will further clarify this point in the paper.
>
> **First statement in the abstract being general:**
> Please note that our results hold for any estimation problem where $X$ is estimated from $Y$, based on a joint distribution $p_{X,Y}$ (see e.g. Theorem 1). $X$ and $Y$ do not necessarily have to be images, and the problem setting is not restricted to image restoration.
>
> **Societal impact:**
> Thanks, we’ll add a discussion on this.

---

> > ### Comment · Reviewer_WGqG · 2021-08-31
> > **Response to rebuttal**
> >
> > I thank the authors for their response. I think they did a good job addressing my concerns. I will raise my score by 1 point, and would expect the authors to add a detailed discussion (and additional theoretical results if possible) for distortions/divergences beyond the MSE/Wasserstein-2 case in the final version of the paper.

---

### Official Review · Reviewer_McgJ · 2021-07-16

**Rating:** 8
**Confidence:** 4

**Summary:**

The authors of this paper suggest that they show four novel contributions to the image compression literature that hold true for MSE distortion and Wasserstein-2 perception index:
1. They prove that the Distortion-perception function is always quadratic in the perception constraint P regardless of the underlying distribution
2.  The show that it is possible to construct estimators of the Distortion-perception curve from estimators at the two extremes of the tradeoff (the one that globally minimizes MSE and the one that minimizes MSE under a perfect perceptual quality constraint).
3.In the gaussian setting, they provide a closed form expression for optimal estimators and the corresponding Distortion-perception curve, they also show that this is a lower bound on the curve of any distribution having the same second-order stats.
4.They illustrate their results conclusively with super-resolution as their test case.

**Main Review:**

I believe that this is a very good paper, that makes novel contributions to an exciting area of the compression field.  To me the biggest contribution of this paper is Theorem 3 (Equation 15) which says that a linear interpolation between the two edge estimators can construct an estimator of the distortion-perception curve that outperforms other estimators at locations along the curve that are not at either extreme.

This is a strong, novel contribution worthy of NeurIPS.

The main weakness of the paper is in the visual comparisons presented, the authors don't show the ground truth image patch with the decoded patches, so even though the results seem to suggest their outcome is better than the comparisons, it is not entirely clear.  In addition, visual comparison at more points along the curve would also be useful.

**Time Spent Reviewing:**

2

---

> ### Author Response · Authors · 2021-08-09
> **Response to Reviewer McgJ**
>
> Thank you for your positive comments. We are gratified that you find the results important and novel.
>
> Regarding your concern about visual comparisons, we kindly refer to Appendix D, where we supply full comparisons (including ground truth patches) and additional points along the D-P curve. These were not included in the main text due to the limited space, but we refer to them from the main text (l. 314). We will highlight this point in the main text.

---

### Official Review · Reviewer_7S5X · 2021-07-24

**Rating:** 5
**Confidence:** 2

**Summary:**

Distortion-perception is the phenomenon that the better a criterion (on natural images) is optimized (for instance SNR in image denoising) the stronger the deviation to the distribution of the space of natural images. This paper characterises this phenomenon under Mean Squared Error (MSE) as the criterion under a Wasserstein bound on the conditional distribution. The result is an interpolation between the distribution of the conditional distribution and the distribution itself. An explicit formulation of the optimizer of their Distortion-perception function is provided. Simulations are made, showing that interpolating different models that approximates the quantities in their theoretical formulation can achieve better visual aspect.

**Limitations And Societal Impact:**

No particular comments.

**Main Review:**

Studying criterion of optimization may appear of increasing interest for the community and it seems to be already the case.

Main concerns: *More background and perspectives are needed in the presentation of the work*

My main concern with the paper is that the technical results are quite straightforward and take too much space in the paper.
Discussing the distortion-perception function in the introduction would have been more informative. In particular, what are the challenges to estimates the quantities involved in this distortion-perception function. I reckon that the Wasserstein framework is nice to obtain this simple interpolation formula which is almost immediate consequence of the definition. So, I would have expected more discussion on why Wasserstein, what about other divergences, why not at least discussing other divergences in order to highlight the different bottlenecks. How to estimate the quantities involved and gentle introduction to these questions would have been a plus.

Understanding the experimental section was difficult, although I am not an expert in this area. Explaining what is done with a more pedagogical presentation would be beneficial.

Other comments:
- About the technical part, I think the authors should not spend so much space of the results and I would suggest to put the proof of the main result in the main text in one theorem and one corollary at most.
- The Gelbrich distance is most often called the Bures-Wasserstein metric. Bures simply is a metric introduced in quantum physics, long before Gelbrich found the closed form although without translation term.


**Time Spent Reviewing:**

3.5

---

> ### Author Response · Authors · 2021-08-09
> **Response to Reviewer 7S5X**
>
> Thank you for your comments.
>
> **Technical results are straightforward and take too much space:**
> First, we beg to differ about the straightforwardness of our results. Note that despite extensive recent research on the topic, our results were never obtained previously, not even for simple special cases. Furthermore, as noted by other reviewers, they are of both conceptual and practical relevance. While the convex combination result (Theorem 3) may seem basic, it was not at all clear from the outset that this result is indeed true. Moreover, the explicit expression we obtained for the general Gaussian case (Theorems 4,5) required extensive matrix analysis and use of properties of optimal transport in Gaussian spaces. Again, we do not view these results as straightforward. As for your concern about space in the text, the proofs of our results extend over more than 7 pages in the supplementary material, and cannot be moved to the main text.
>
> **More background:**
> We’ll expand the discussion on the general perception-distortion function in the introduction,  and place our results in context.
>
> **Why Wasserstein and discussion of other divergences:**
> As opposed to most divergences (like KL and other f-divergences), Wasserstein distance can be evaluated and optimized for singular distributions (i.e. not absolutely continuous w.r.t. each other). As images lie on low-dimensional manifolds, Waserstein-GANs (WGANs) made a tremendous impact, and practically are predominantly used in current generative models (e.g. BigGAN, StyleGAN, SinGAN, etc.).
>
> Regarding divergences other than Wasserstein-2, our results can be used to derive bounds on the D-P function with other divergences. In particular, (8) yields lower bounds on D(P) w.r.t. Wasserstein-p (for p>2), $W_1$ (with slight modifications), KL (when $p_X$ is Gaussian). Needless to say, at the point P=0 , D(P=0) coincides with (8) for all divergences.  We will add this discussion about other divergences (with MSE distortion) to the text.Thanks for your remark.
>
> **How to estimate quantities:**
>
> *Estimating Wasserstein distances:* as they are extensively used in practice, many standard and improved methods of evaluation can be found in the literature (see e.g. [R1,R2,R3] below). We will add references to common practice and add a discussion to the text.
>
> *Approximating the estimators* ($\hat{X}_0, X^*$, etc.): We discuss some common practices, see paragraph starting at l. 131. We will expand the discussion on this.
>
> [R1] Arjovsky et al. “Wasserstein GAN”, International conference on machine learning, 2017.
>
> [R2] Gulrajani, et al. “Improved training of wasserstein gans”, Neurips 2017.
>
> [R3] Leygonie et al. “Adversarial Computation of Optimal Transport Maps”,  arXiv preprint arXiv:1906.09691, 2019.
>
> **Better explanation of experiments:**
> Thanks for the comment, we will explain the experimental illustration for the broader audience in a more detailed and visual manner in the Appendix. We will visualize the SR problem setting - what are $X,Y,\hat X$ etc. with a concrete example.
>
> **Bures distance:**
> Thanks for the pointer. We will mention the Bures-distance in the context of the Gelbrich distance.

---

### Decision · Program_Chairs · 2021-09-27

**Decision:**

Accept (Poster)

**Comment:**

Reviewers found this paper to provide an interesting and welcome application for optimal transport, even if the theoretical contribution is limited from a technical perspective.  After some deliberation, we converged on accepting this paper.

In the revision, please provide the following:

* More experimental evidence corroborating that a convex combination of two estimators results in visually pleasing results.
* Better introduction to the perception-distortion tradeoff for non-experts
* Optionally:  Detailed discussion (and additional theoretical results if possible) for distortions/divergences beyond the MSE/Wasserstein-2 case.